# Metabolomic profiling reveals a differential role for hippocampal glutathione reductase in infantile memory formation

**Benjamin Bessières, Emmanuel Cruz, Cristina M Alberini***

Center for Neural Science, New York University, New York, United States

**Abstract** The metabolic mechanisms underlying the formation of early-life episodic memories remain poorly characterized. Here, we assessed the metabolomic profile of the rat hippocampus at different developmental ages both at baseline and following episodic learning. We report that the hippocampal metabolome significantly changes over developmental ages and that learning regulates differential arrays of metabolites according to age. The infant hippocampus had the largest number of significant changes following learning, with downregulation of 54 metabolites. Of those, a large proportion was associated with the glutathione-mediated cellular defenses against oxidative stress. Further biochemical, molecular, and behavioral assessments revealed that infantile learning evokes a rapid and persistent increase in the activity of neuronal glutathione reductase, the enzyme that regenerates reduced glutathione from its oxidized form. Inhibition of glutathione reductase selectively impaired long-term memory formation in infant but not in juvenile and adult rats, confirming its age-specific role. Thus, metabolomic profiling revealed that the hippocampal glutathione-mediated antioxidant pathway is differentially required for the formation of infantile memory.

## Editor's evaluation

In this study the authors examined the metabolomic profile of the rat hippocampus and report significant changes at various developmental ages at baseline and following episodic learning. Infants had the largest number of changes of hippocampal metabolites with many associated with the glutathione mediated antioxidant pathway. Infantile learning was shown to induce a rapid increase in glutathione reductase and inhibition of this enzyme impaired long term memory formation at this developmental age, but not in older animals, suggesting a key requirement for this pathway in infantile memory formation.

**\*For correspondence:**
ca60@nyu.edu (CMA);
ca60@nyu.edu (CMA)

## Introduction

The hippocampus and related brain structures of the medial temporal lobe, which play critical roles in the formation of spatial and episodic memories (*Squire et al., 2004*), undergo a protracted postnatal developmental during which significant molecular, cellular, and structural changes accompany circuit maturation and refinement (*Bandeira et al., 2009*; *Lavenex and Banta Lavenex, 2013*; *Albani et al., 2014*; *Travaglia et al., 2016a*). This maturation, which is thought to be guided by early-life experiences (*Travaglia et al., 2016b*; *Bessières et al., 2020*), leads to the acquisition of adult-like abilities to form and express long-lasting memories (*Dumas, 2005*; *Newcombe et al., 2007*; *Rovee-Collier and Cuevas, 2009*; *Mullally and Maguire, 2014*; *Donato et al., 2021*).

Unlike the memories formed when the system is mature, episodic memories learned in infancy are expressed for a short time, but then appear to be rapidly forgotten. This forgetting is thought to be associated to infantile amnesia, the inability of adults to recall early-life events (*Campbell and Spear, 1972*; *Hayne, 2004*; *Josselyn and Frankland, 2012*; *Callaghan et al., 2014*; *Madsen and Kim, 2016*; *Alberini and Travaglia, 2017*). However, infantile memories are not lost, but rather are stored over the long term in a latent form; in fact, they can be reinstated later in life following the presentation of certain behavioral reminders or artificial reactivation of the neuronal networks activated during learning (*Travaglia et al., 2016b*; *Guskjolen et al., 2018*; *Bessières et al., 2020*).

Compared to juvenile or adult learning, which result in long-term memory expression, infantile learning engages unique molecular mechanisms in the dorsal hippocampus (dHC), including a mGluR5 and BDNF-dependent switch in the expression of *N*-methyl-D-aspartate (NMDA) receptor subunits GluN2A/GluN2B; a persistent increase in the immediate early genes (IEGs) c-Fos, Zif268, and activity-regulated cytoskeleton-associated protein (Arc/Arg3.1); an elevated expression of excitatory synapse markers synaptophysin and postsynaptic density 95 (PSD-95); and the maturation of α-amino-3-hydroxy-5-methyl-4-isox-azoleproprionic acid (AMPA) receptor synaptic responses (*Travaglia et al., 2016b*; *Bessières et al., 2020*). All these changes, which contribute to the maturation of the hippocampal memory system and acquisition of the adult-like ability to express memory over the long term (*Travaglia et al., 2016b*; *Bessières et al., 2020*), consume large amounts of energy and consequently require a high level of energy metabolism regulations.

At baseline, the developing brain consumes higher levels of energy than the adult brain (*Kuzawa et al., 2014*). Furthermore, the utilization of energy substrates changes both across developmental ages (*Cremer, 1982*; *Nehlig, 2004*; *Brekke et al., 2015*) and in response to activity-dependent processes such as learning and long-term plasticity (*Suzuki et al., 2011*; *Bélanger et al., 2011*; *Alberini et al., 2018*), suggesting that the brain might differentially regulate energy metabolism in response to learning across developmental ages. Elucidating such differential learning-dependent changes would improve our understanding of the distinctive biology of the brain across developmental ages and provide key insights into the pathogenesis and treatment of brain diseases at different ages.

Although a few studies have reported that the rat hippocampal metabolome changes across the lifespan (*Zheng et al., 2016*), as well as during aging (*Zhang et al., 2009*), little is known about how the hippocampal metabolomic profile changes across development and in response to learning at different ages. To fill this knowledge gap, we employed an untargeted metabolomic analyses in rats to determine how the hippocampal metabolome changes over the course of postnatal development under basal conditions and following inhibitory avoidance (IA) training, an aversive episodic event. We found that unique metabolomic profiles accompany learning at different ages. Subsequent biochemical, pharmacological, and behavioral studies based on unique metabolomic regulations in the infant hippocampus established that infantile learning selectively recruits the glutathione-mediated antioxidant defenses for the formation of long-term memory.

## Results and discussion

### The metabolic profile of the hippocampus significantly changes across developmental ages

Metabolomic profiling was carried out on hippocampal extracts (five to seven individual replicates) obtained from untrained rats at four early prepuberal developmental ages (postnatal day 1 [PN1], PN7, PN17, and PN24) and compared to the profile obtained from young adult (PN80) rats. These ages were chosen because they correspond to critical stages in the functional development of the hippocampal memory system: at PN1 and PN7, rats cannot perform IA; at PN17, they form short-term memories that are rapidly forgotten; and at both PN24 and PN80, they form and express strong and long-lasting IA memories (*Travaglia et al., 2016a*; *Travaglia et al., 2016b*; *Bessières et al., 2020*).

The metabolomic profiling of the hippocampi was preformed using an untargeted ultrahigh performance liquid chromatography-tandem mass spectroscopy (UPLC-MS/MS). Identification of metabolites was based on a comparison of the retention time/index (RI), the mass-to-charge ratio (m/z) +/10 ppm, and the MS/MS forward and reverse scores between metabolites detected in the hippocampal extracts and more than 3300 purified standard compounds maintained in a reference library compiled by Metabolon Inc (Morrisville, NC). This approach led to the detection and identification of a total of

454 metabolites across all samples, which belonged to eight distinct pathways (*Figure 1a*): 145 amino acids, 50 carbohydrates, 31 cofactors and vitamins, 11 metabolites involved in energy metabolism, 110 lipids, 52 nucleotides, 38 peptides, and 17 xenobiotics.

To obtain the metabolomic profile comparisons at baseline among the different ages, we first employed a principal component analysis (PCA), an unsupervised classification method that allows the reduction of the dimensionality of the dataset while preserving most of the variance between samples (*Worley and Powers, 2013*). As shown in *Figure 1b*, PCA revealed that all samples clustered without overlap into five distinct populations, which corresponded to the five developmental ages. This analysis revealed that the metabolomic profiles were homogenous among samples from each age group, whereas samples of different ages differed dramatically, suggesting that the rat hippocampus undergoes major metabolic changes over the course of development. No separation of the samples between males and females was observed following PCA (*Figure 1—figure supplement 1*), indicating that there was no significant sex-related difference in the dataset (see Materials and methods section for additional information).

Next, to better characterize the changes in the hippocampal metabolome over development, we compared the total peak number, as a measure of the total number of metabolites detected at each age, and the total peak intensity (following median scaling) of the identified metabolites across ages (*Figure 1c*). Total peak number and scaled intensity were similar between PN1 and PN7, but then significantly increased between PN7 and PN17, increased further between PN17 and PN24, and then significantly decreased between PN24 and PN80. To assess the temporal progression of the hippocampal metabolome over the course of the hippocampus maturation, we compared the total number of metabolites significantly upregulated and downregulated at each age compared to the previous age (i.e., PN7 vs. PN1, PN17 vs. PN7, PN24 vs. PN17, and PN80 vs. PN24) (*Figure 1d*). From PN1 to PN7, 49.6% of the detected metabolites changed significantly, and changes were similarly distributed between up- and downregulations. Comparisons of PN17 to PN7 and PN24 to PN17 revealed significant changes in a large number of metabolites (62.78% between PN7 and PN17; 63.66% between PN17 and PN24), the majority of which were upregulated. From PN24 to PN80, 72.03% of the metabolites changed significantly, and the great majority of them were significantly downregulated. Collectively, these results indicated that the metabolic profile of the hippocampus changes significantly across developmental ages, with the diversity and concentration of metabolites peaking at PN24 before declining in adulthood.

## Levels of amino acids and nucleotides decrease, whereas levels of lipids and cofactors/vitamins increase, over the course of postnatal hippocampal development

We next examined the age-related progression of metabolic pathways, as defined in *Figure 2*. To this end, we calculated the mean peak intensities of the metabolites in each pathway at each age and compared them with those in adult rats (PN80), using unsupervised hierarchical cluster analysis (HCA) based on the $\log_2$(fold change) (*Figure 2*). Most of the pathways belonging to the amino acid and nucleotide metabolisms, along with the lipid pathways *ketone bodies* and *glycerolipid and fatty acids*, and the carbohydrate pathways *glycogen metabolism, nucleotide sugars*, and *aminosugar metabolism* were elevated at PN1 and PN7 relative to PN80. By contrast, the pathways that were significantly lower in early development relative to PN80 belonged mainly to lipid and cofactor/vitamin metabolisms (with the exception of the lipid pathways *ketone bodies* and *glycerolipid and fatty acids*) and to four carbohydrate and energy pathways: *advanced glycation end products, TCA cycle, pentose phosphate pathway*, and *oxidative phosphorylation*. Overall, these developmental metabolic changes may reflect the increase in glial cell population and the neuronal circuit maturation accompanying postnatal hippocampal growth (*Bandeira et al., 2009*; *Lavenex and Banta Lavenex, 2013*; *Albani et al., 2014*).

To determine which metabolites changed more significantly across ages, we performed a random forest (RF) analysis (*Grissa et al., 2016*; *Figure 3a*). RF is a classification algorithm that uses an ensemble of decision trees, which allow selection of metabolites based on their ability to decrease the classification accuracy of the algorithm following their permutation among sample groups (*Grissa et al., 2016*). As shown in *Figure 3a*, the RF analysis clustered the samples into five groups, which corresponded to the five ages (PN1, PN7, PN17, PN24, and PN80), with a predictive

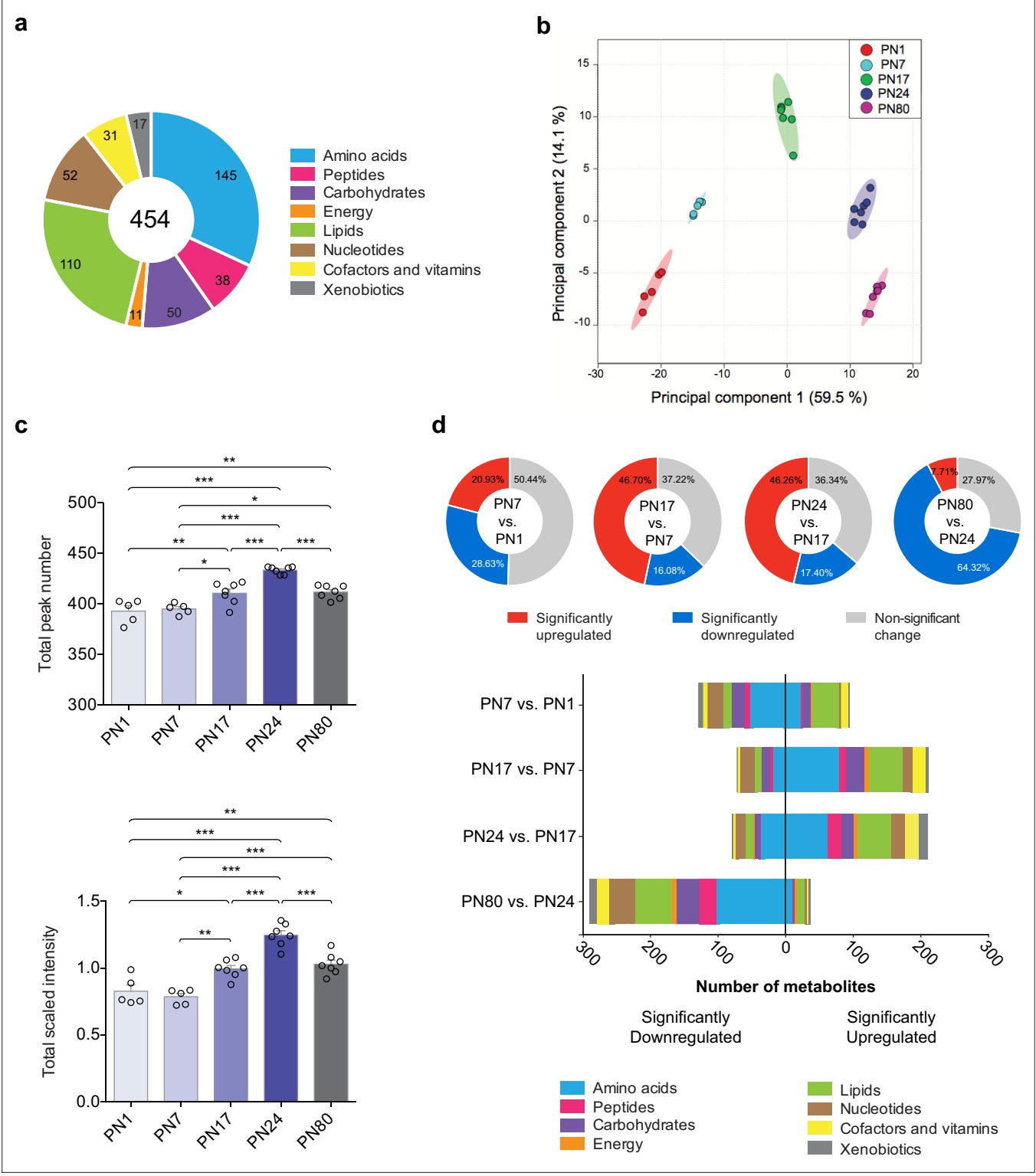

**Figure 1.** Overview of hippocampal metabolomic changes across postnatal developmental ages. (**a**) Pie chart showing the total number (454) of metabolites identified in the whole rat hippocampus across developmental ages by untargeted ultrahigh performance liquid chromatography-tandem mass spectroscopy (UPLC-MS/MS). The metabolites belonged to eight different pathways: amino acids (145), carbohydrates (50), cofactors and vitamins (31), energy metabolism (11), lipids (110), nucleotides (52), peptides (38), and xenobiotics (17). (**b**) Principal component analysis (PCA) of untrained groups at different developmental ages: postnatal day 1 (PN1) (n = 5), PN7 (n = 5), PN17 (n = 7), PN24 (n = 7), and PN80 (n = 7). The PCA plot revealed a substantial separation of samples based on age; each dot represents a single sample. The samples distributed into five distinctive populations (without

*Figure 1 continued on next page*

*Figure 1 continued*

overlap) corresponding to the five developmental ages. Colored ellipses in the score plot depict 95% confidence areas. The first and second principal components explained 59.5% and 14.1% of the total variance, respectively. (**c**) Mean values ± s.e.m. of total numbers of identified metabolites and total peak intensities, after median scaling, of the untrained groups at each developmental age. *p < 0.05, **p < 0.01, ***p < 0.001 (one-way ANOVA followed by Tukey's multiple comparisons test). (**d**) Total number of metabolites significantly upregulated and downregulated at each age in comparison with the previous age (PN7 vs. PN1, PN17 vs. PN7, PN24 vs. PN17, and PN80 vs. PN24) and distribution of the metabolites per pathway. Statistical significance: p < 0.05, q-value < 0.1 (one-way ANOVA followed by Tukey's multiple comparisons test and estimation of the false discovery rate [FDR]); one experiment. See *Figure 1—source data 1* for numerical values and detailed statistical information.

The online version of this article includes the following source data and figure supplement(s) for figure 1:

**Source data 1.** Numerical values and detailed statistical information.

**Figure supplement 1.** Principal component analysis (PCA) reveals no sex difference within the naive and trained groups.

**Figure supplement 1—source data 1.** Numerical values and detailed statistical information.

accuracy of 100% (random classification would have given a predictive accuracy of 20%). These results confirmed that the metabolomic profile of the hippocampus is substantially distinct among the age groups. Then, variable permutations and measures of the decrease in classification accuracy were used to identify the 30 top-ranked metabolites that make the largest contribution to the classification. The 30 top-ranked metabolites included 12 amino acids, 3 carbohydrates, 1 cofactor or vitamin, 2 metabolites involved in energy metabolism, 6 lipids, and 6 nucleotides (*Figure 3a*). Notably, 5 out of the 12 amino acids belonged to the urea cycle, arginine, and proline metabolism (homoarginine, *N*-delta-acetylornithine, trans-4-hydroxyproline, *N*-acetylarginine, dimethylarginine) and 3 to histidine metabolism (1-ribosyl-imidazoleacetate, imidazole propionate, imidazole lactate). Four lipids out of six belonged to carnitine/acylcarnitine metabolism: arachidonoylcarnitine (C20:4), 3-hydroxybutyrylcarnitine, linoleoylcarnitine (C18:2), and myristoylcarnitine (C14). Finally, the six top-ranked nucleotides belonged in equal proportions to purine (adenine, guanine, allantoin) and pyrimidine (5'-CMP, orotidine, cytosine) metabolism (*Figure 3a*).

We next assessed how the 30 top-ranked metabolites identified by RF analysis (*Figure 3a*) changed across ages relative to PN80, using an HCA based on the $\log_2$(fold change) (*Figure 3b*). Consistent with the developmental changes of the main metabolic pathways (*Figure 2*), the concentrations of most of the top-ranked amino acids and nucleotides were higher at early postnatal developmental ages than in adulthood, whereas the concentrations of the majority of the top-ranked lipids were lower at early postnatal development ages compared to adults. Moreover, most of these 30 top-ranked metabolites peaked at PN17 or PN24 (*Figure 3b*).

The top-ranked metabolite that changed across developmental ages was the sphingoid base *sphinganine* (*Figure 3a and b*), which forms the backbone of all sphingolipids (*Carreira et al., 2019*). Sphingolipids are important second messengers that play roles in the regulation of cell survival and proliferation, cell growth, cell differentiation, cell migration, apoptosis, angiogenesis, and endothelial barrier integrity (*Soriano et al., 2005*), as well as in myelination (*Schmitt et al., 2015*). Myelin is a spiral extension of the oligodendroglial cell membrane that increases axonal conduction velocity and efficiency, provides additional metabolic support to axons, and stabilizes axonal projections in the developing and adult mammalian brain (*Nave, 2010*). Myelin contains a very high lipid-to-protein ratio, and formation of the myelin sheath requires a high level of lipid synthesis, which is temporally regulated during brain development (*Schmitt et al., 2015*). In the rat hippocampus, mature myelinated fibers appear during the second postnatal week, and their distribution reaches the adult level at PN25 (*Meier et al., 2004*; *van Tilborg et al., 2018*). Consistent with the correlation between sphinganine and the myelination rate during development, we found that the level of sphinganine (*Figure 3b*), along with those of other sphingolipids (*Figure 2*), were much lower at PN1 than at PN80. Sphinganine level gradually increased over postnatal development, peaking at PN24, when the myelination rates in the rodent brain are the highest (*Meier et al., 2004*; *van Tilborg et al., 2018*).

Among the top-ranked metabolites were also several related to histidine metabolism: 1-ribosyl-imidazole acetate, imidazole propionate, and imidazole lactate. Imidazole acetate and its derivatives are the end products of the transformation of histamine, which is produced by decarboxylation of histidine, and can act as $GABA_A$ agonists (*Glinton et al., 2019*). Imidazole propionate and imidazole lactate are also produced from histidine and can be used as precursors for the synthesis of glutamate (*Glinton et al., 2019*). Therefore, the increase in histidine-related metabolism during

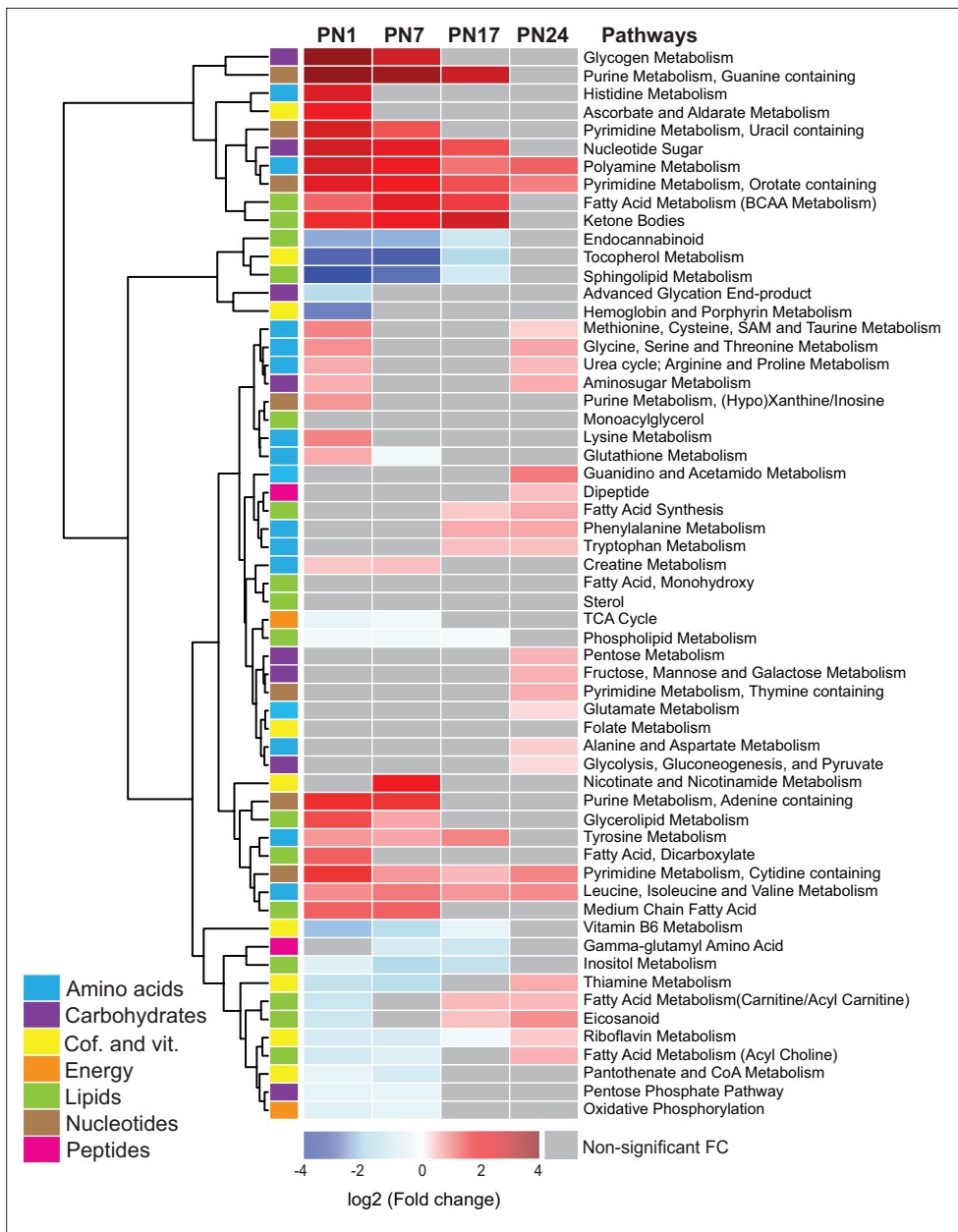

**Figure 2.** Changes in the main metabolic pathways across development compared to adult age. Unsupervised hierarchical cluster analysis (HCA) of 59 metabolomic pathways grouped into seven main endogenous families of metabolites: amino acids, carbohydrates, cofactors and vitamins, energy metabolism, lipids, nucleotides, and peptides. HCA was generated using Euclidean distances based on the $log_2$(fold change) values of mean peak intensities associated with each pathway at each age, relative to the corresponding values at postnatal day 80 (PN80). Red and blue boxes in the heatmap indicate significant positive and negative fold changes, respectively. Gray boxes indicate non-significant fold changes (FC). In the dendrogram, metabolomic pathways that undergo similar variations across development are clustered close to one another. Statistical significance: $p < 0.05$ (one-way ANOVA followed by Tukey's multiple comparisons test). n = 5–7 rats per group; one experiment. See *Figure 2— source data 1* for numerical values and detailed statistical information.

The online version of this article includes the following figure supplement(s) for figure 2:

**Source data 1.** Numerical values and detailed statistical information.

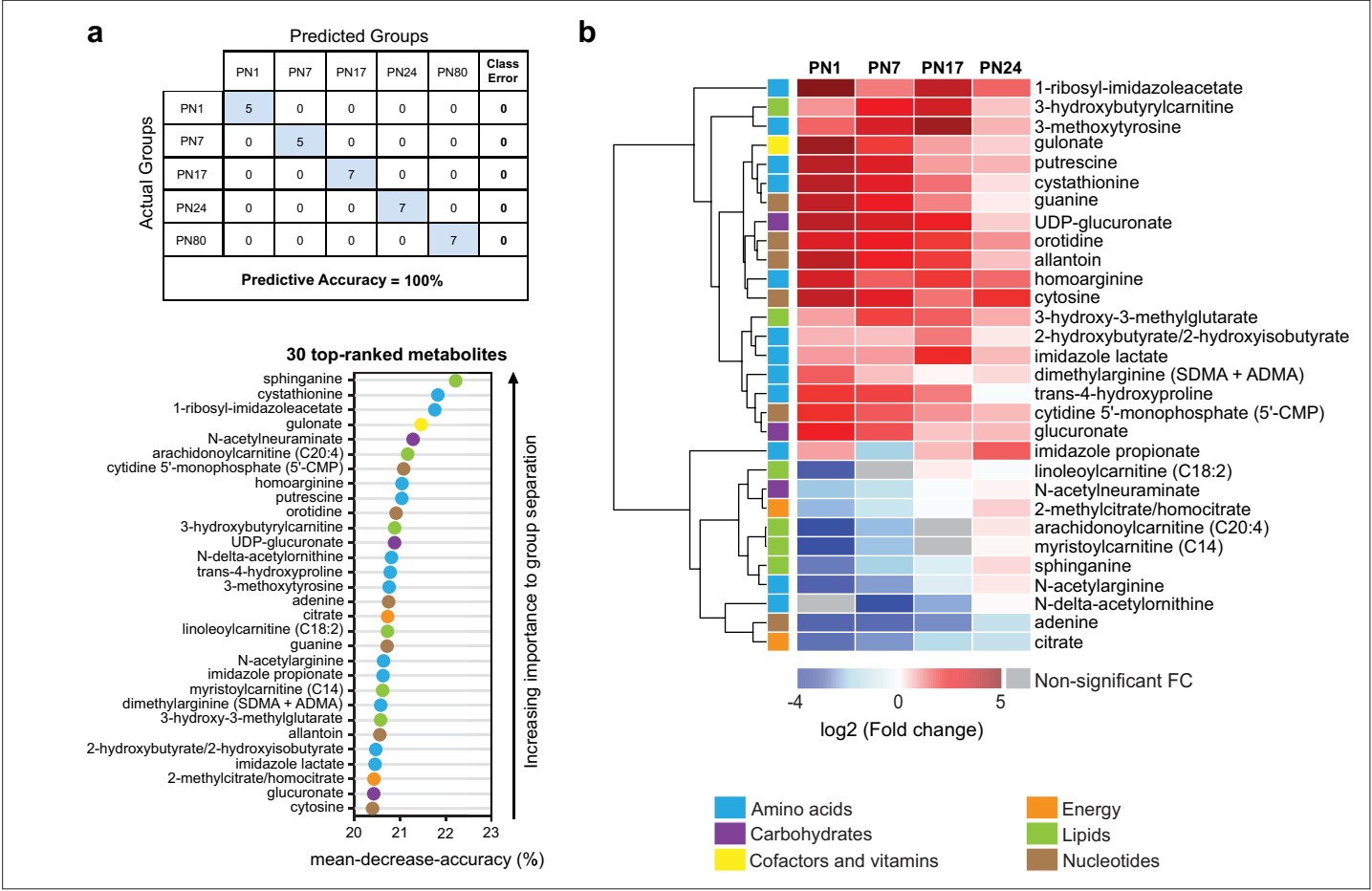

**Figure 3.** The 30 top-ranked metabolites that change in the hippocampus over post-natal development. (**a**) Random forest (RF) analysis was used to identify the 30 top-ranked metabolites that contribute to the differences between the five untrained developmental groups. Variable permutations and measures of 'mean decrease accuracy' were used to identify the 30 top-ranked metabolites that make the largest contribution to the classification. As shown in the table, RF analysis yielded a predictive accuracy of 100% (random segregation of samples would give a predictive accuracy of 20%), confirming accurate classification of the samples into the five developmental groups. (**b**) Unsupervised hierarchical cluster analysis (HCA) of the 30 top-ranked metabolites identified after RF analysis. HCA was generated using Euclidean distances based on the $\log_2$(fold change) values of mean peak intensities associated with each metabolite at each age relative to the corresponding values at postnatal day 80 (PN80). Red and blue boxes in the heatmap indicate significant relative increases and decreases in metabolite levels, respectively. Gray boxes indicate non-significant fold change (FC) values. In the dendrogram, metabolites that undergo similar variations across development are clustered close to one another. Statistical significance: $p < 0.05$, q-value $< 0.1$ (one-way ANOVA followed by Tukey's multiple comparisons test and estimation of the false discovery rate [FDR]). n = 5–7 rats per group; one experiment. See *Figure 3—source data 1* for numerical values and detailed statistical information.

The online version of this article includes the following figure supplement(s) for figure 3:

**Source data 1.** Numerical values and detailed statistical information.

development may reflect higher glutamate synthesis in juvenile rats. Consistent with this hypothesis, the hippocampal levels of glutamate and other biochemicals involved in glutamate metabolism, including aspartate, glutamine, and GABA, peaked at PN24.

Moreover, four of the six lipids were associated with carnitine/acyl carnitine metabolism: arachidonoylcarnitine (C20:4), 3-hydroxybutyrylcarnitine, linoleoylcarnitine (C18:2), and myristoylcarnitine (C14); all peaked between PN17 and PN24 (*Figure 3b*). Carnitine and its acyl derivatives are important transporters of long-chain fatty acids into mitochondria for β-oxidation (*Jones et al., 2010*). Thus, the increase in the level of acyl carnitines during the first 3 weeks of life may indicate that the level of fatty acid β-oxidation is higher in the juvenile brain than in adulthood. In line with this hypothesis, previous studies revealed the ability of the developing brain to use ketone bodies through β-oxidation, as a complementary source of energy to sustain the dramatic metabolic demands of the brain during its development (*Cremer, 1982*; *Dombrowski et al., 1989*; *Nehlig, 2004*). Finally, the decrease in

nucleotide levels between PN1 and adulthood (aside from adenine, for which the level increased gradually from PN1 to PN24) may reflect a reduction in DNA and RNA synthesis, and hence a decrease in cell proliferation, over the course of postnatal development.

## Significant changes in the levels of carbohydrates and tricarboxylic acid cycle metabolites over development

Carbohydrates and tricarboxylic acid (TCA) cycle-related metabolites function as major metabolic substrates to fuel multiple cellular functions, including synthesis of neurotransmitters, nucleic acids, proteins, and lipids, needed for cell and axonal growth, myelination, and synapse formation and maturation (*Brekke et al., 2015*). We assessed how the hippocampal levels of these metabolic energy substrates changed at each age relative to PN80, using nine separated HCAs, one for each sub-pathway: (1) glycolysis, gluconeogenesis, and pyruvate metabolism, (2) pentose phosphate pathway, (3) pentose metabolism, (4) fructose, mannose, and galactose metabolism, (5) nucleotide sugar, (6) aminosugar, (7) TCA cycle, (8) oxidative phosphorylation, and (9) glycogen metabolism (*Figure 4*). As previously described, the HCAs were generated based on the $\log_2$(fold change) of mean peak intensities associated with each metabolite at each age relative to PN80 (*Figure 4*). We found that the concentration of metabolites related to glycogen (i.e., maltotetraose, maltotriose, and maltose) and nucleotide sugar (e.g., UDP-sugar) metabolisms was substantially higher during postnatal development, especially early after birth (PN1–PN7), than in adulthood, suggesting higher glycogen, glycolytic metabolism, and nucleotide synthesis early in life. Most of the other carbohydrates and TCA cycle-related metabolites identified in our study peaked at PN17 or PN24.

Metabolites related to glycolysis followed different profiles across development (*Figure 4*). Glucose, glucose-6P, pyruvate, and lactate peaked at PN17 or PN24. Fructose 1,6-bisP and dihydroxy-acetone phosphate gradually increased with age from PN1 to PN80, whereas 3-phosphoglycerate and phosphoenolpyruvate underwent a progressive decrease across development. Four metabolites out of seven belonging to the pentose phosphate pathway, PRPP, xylulose-5P, ribose-5P, and sedoheptulose-7P, reached a plateau at PN17 or PN24 (*Figure 4*). Finally, within the TCA cycle, citrate and aconitate gradually increased between PN1 and PN80; alpha-ketoglutarate and succinylcarnitine remained higher during prepuberal ages compared to PN80; fumarate and malate reached peaks at PN24 before declining in adulthood. Collectively, these changes in carbohydrates and TCA cycle-related metabolites suggest that oxidative metabolism peaks at PN17 and PN24, a period during which hippocampus-dependent memories fully develop, acquiring adult-like abilities.

## Episodic learning at different ages differentially regulates the hippocampal metabolome

We next compared the metabolomic profiles of the hippocampus 1 hr after IA training at PN17, PN24, and PN80 with those of age-matched untrained (naive) control rats.

Training at PN17 led to significant downregulation of 54 metabolites: 23 amino acids, 11 cofactors or vitamins, 8 peptides, 5 lipids, 3 nucleotides, 2 carbohydrates, and 2 energy-related metabolites (*Figure 5a and d*). By contrast, training at PN24 did not cause any significant change (*Figure 5b*). At PN80 (*Figure 5c and e*), 9 metabolites (2 amino acids, 1 peptide, 5 lipids, and 1 nucleotide) were significantly upregulated by learning, whereas only 2 carbohydrates emerged as significantly downregulated by comparison with the untrained rats.

Notably, following learning in infancy, the top-ranked downregulated amino acids, cofactors, and vitamins included several molecules involved in ascorbate and glutathione metabolism. In fact, the level of the oxidized form of glutathione (GSSG) significantly decreased after IA training at PN17, along with that of cysteine, a rate-limiting substrate in the biosynthesis of the reduced form of glutathione (GSH) and five other major metabolites involved in cellular defense against oxidative stress: *S*-lactoylglutathione, the methyl donor *S*-adenosylmethionine (SAM), 5-oxoproline, ascorbate, and alpha-tocopherol (*Figure 5a and d*). Despite cysteine decreases at 1 hr following learning, at this time point we did not detect significant change in GSH level (*Figure 5a and d*). Therefore, we speculate that the decrease in cysteine may not be linked to GSH biosynthesis but rather serves as a substrate for de novo protein synthesis, a process known to be required for long-term memory formation (*Costa-Mattioli et al., 2009*). None of these changes were observed at PN24 and PN80 (*Figure 5b and c*), indicating that infantile learning leads to differential activation of the cellular antioxidant defense.

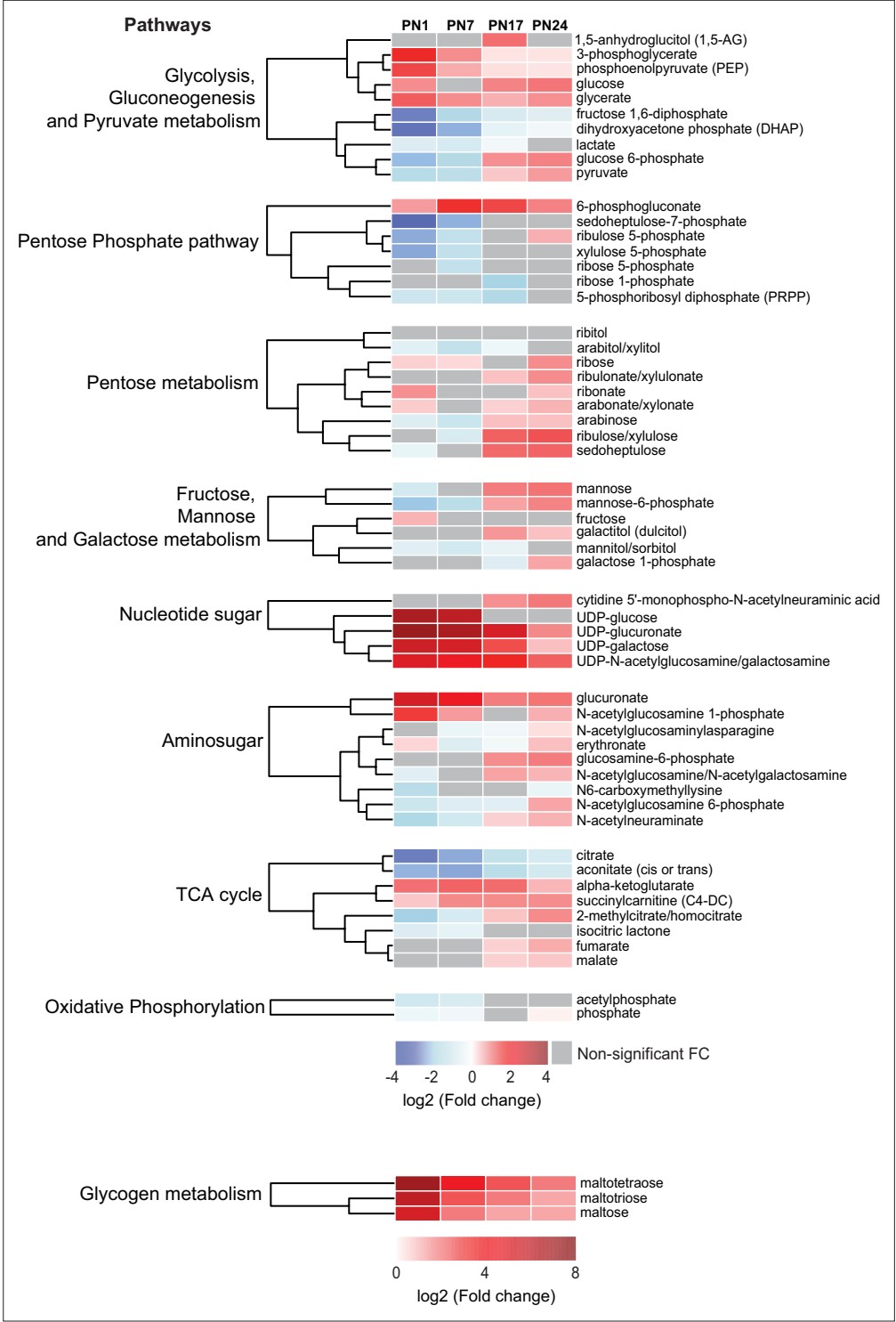

**Figure 4.** Changes in carbohydrates and tricarboxylic acid (TCA) cycle-related metabolites across development. Unsupervised hierarchical cluster analysis (HCA) of 59 metabolites grouped into nine pathways: (1) glycolysis, gluconeogenesis, and pyruvate metabolism, (2) pentose phosphate pathway, (3) pentose metabolism, (4) fructose, mannose, and galactose metabolism, (5) nucleotide sugar, (6) aminosugar, (7) TCA cycle, (8) oxidative phosphorylation, and (9) glycogen metabolism. HCAs were generated using Euclidean distances based on the log$_2$(fold change [FC]) values of mean peak intensities associated with each metabolite at each age relative to the corresponding values at postnatal day 80 (PN80). Red and blue boxes in the heatmap indicate significant relative increases and decreases in metabolite levels, respectively. Gray boxes indicate non-significant FC. In the

*Figure 4 continued on next page*

*Figure 4 continued*

dendrograms, metabolites that undergo similar variations across development are clustered close to one another. Statistical significance: p < 0.05, q-value <0.1 (one-way ANOVA followed by Tukey's multiple comparisons test and estimation of the false discovery rate [FDR]). n = 5–7 rats per group; one experiment. See ***Figure 4—source data 1*** for numerical values and detailed statistical information.

The online version of this article includes the following figure supplement(s) for figure 4:

**Source data 1.** Numerical values and detailed statistical information.

The reasons why these antioxidant defense pathways are recruited only following infantile learning is unclear. We speculate that this recruitment could result from higher oxidative stress induced by higher metabolic activity underlying memory formation in infancy. Indeed, neuronal electrical activity is known to raise ATP demands, which must be met by increased metabolic activity, particularly oxidative phosphorylation (***Harris et al., 2012***; ***Fernandez-Fernandez et al., 2012***; ***Bolaños, 2016***), which results in an increased production of reactive oxygen species (ROS) by the mitochondrial respiratory chain, potentially leading to oxidative stress (***Hongpaisan et al., 2004***; ***Brennan et al., 2009***; ***Frisard and Ravussin, 2006***; ***Quijano et al., 2016***; ***Cobley et al., 2018***). At any age, memory formation requires a great deal of energy as it involves, in addition to neuronal neurotransmission (***Attwell and Laughlin, 2001***), numerous cellular and molecular mechanisms in many cell types, which cooperate in producing the changes needed for memory consolidation and storage. One of these mechanisms is de novo protein synthesis, one of the most energy-costly processes (***Buttgereit and Brand, 1995***). Given the fact that infantile learning engages distinct biological mechanisms, which promote developmental maturation of the brain (***Travaglia et al., 2016a***; ***Bessières et al., 2020***), we suggest that the high antioxidant response observed in the infant brain following learning may result from both a higher energy requirement necessary to promote the developmental maturation and an insufficient antioxidant capacity to contrast the consequent high level of oxidative stress induced by the increased metabolic demands. However, this explanation may be contrasted by a recent study showing that the mitochondrial respiration rate (i.e. metabolic activity) in neurons and astrocytes is inversely correlated to ROS production, hence potential oxidative stress (***Lopez-Fabuel et al., 2016***). Furthermore, it is likely that in order to promote developmental maturation, infantile learning recruits more energy compared to the adult, hence it differentially engages metabolic mechanisms in different cell types. In fact, adult rodents forming hippocampus-dependent memories recruit hippocampal astrocytes, which mostly rely on glycolysis, to provide energy to neurons (***Suzuki et al., 2011***; ***Newman et al., 2011***). Therefore, the higher energy demands of the infant brain supporting learning may over solicit astrocytes, which, compared to neurons, exhibit a lower mitochondrial respiration rate leading to a higher ROS production (***Lopez-Fabuel et al., 2016***). Future studies should assess the contributions of astrocytic-neuronal metabolic coupling associated to hippocampus-dependent memory formation in infants relative to adults.

Significant alterations in amino acid levels were also detected following training at PN17 and not at PN24, with 11 of 20 proteinogenic amino acids exhibiting statistically significant decreases (cysteine, asparagine, valine, alanine, tryptophan, glutamine, lysine, histidine, proline, serine, glutamate). One likely cause of these decreases is that these amino acids are rapidly consumed by learning-induced translation (***Salinas et al., 1990***). It is also possible that, because infantile learning is more energy-demanding than learning at later ages, amino acids are used as an energy source. Infantile training also resulted in changes in glycolytic and TCA cycle intermediates, particularly in decreases in lactate, erythronate, fumarate, and malate implying that learning leads to a higher metabolic rate in the TCA cycle.

In addition, glutamate and gamma aminobutyric acid (GABA), as well as biochemicals involved in glutamate/glutamine/GABA metabolism, including asparagine, glutamine, *N*-acetylaspartate, and *N*-acetyl-aspartyl-glutamate (NAAG), significantly decreased following training at PN17, suggesting that the hippocampus of infant rats rapidly consumes these neurotransmitters in response to learning.

On the other hand, as shown in ***Figure 5e***, the top three biochemicals upregulated by learning in adulthood were histidine-derived metabolites (1-methyl-4-imidazoleacetate, imidazole propionate, and gamma-glutamylhistidine). As previously mentioned, imidazole acetate and its derivatives can act as GABA$_A$ agonists (***Glinton et al., 2019***). Imidazole propionate is also produced from histidine and is used as precursor for the synthesis of glutamate (***Glinton et al., 2019***). Thus, the increase in

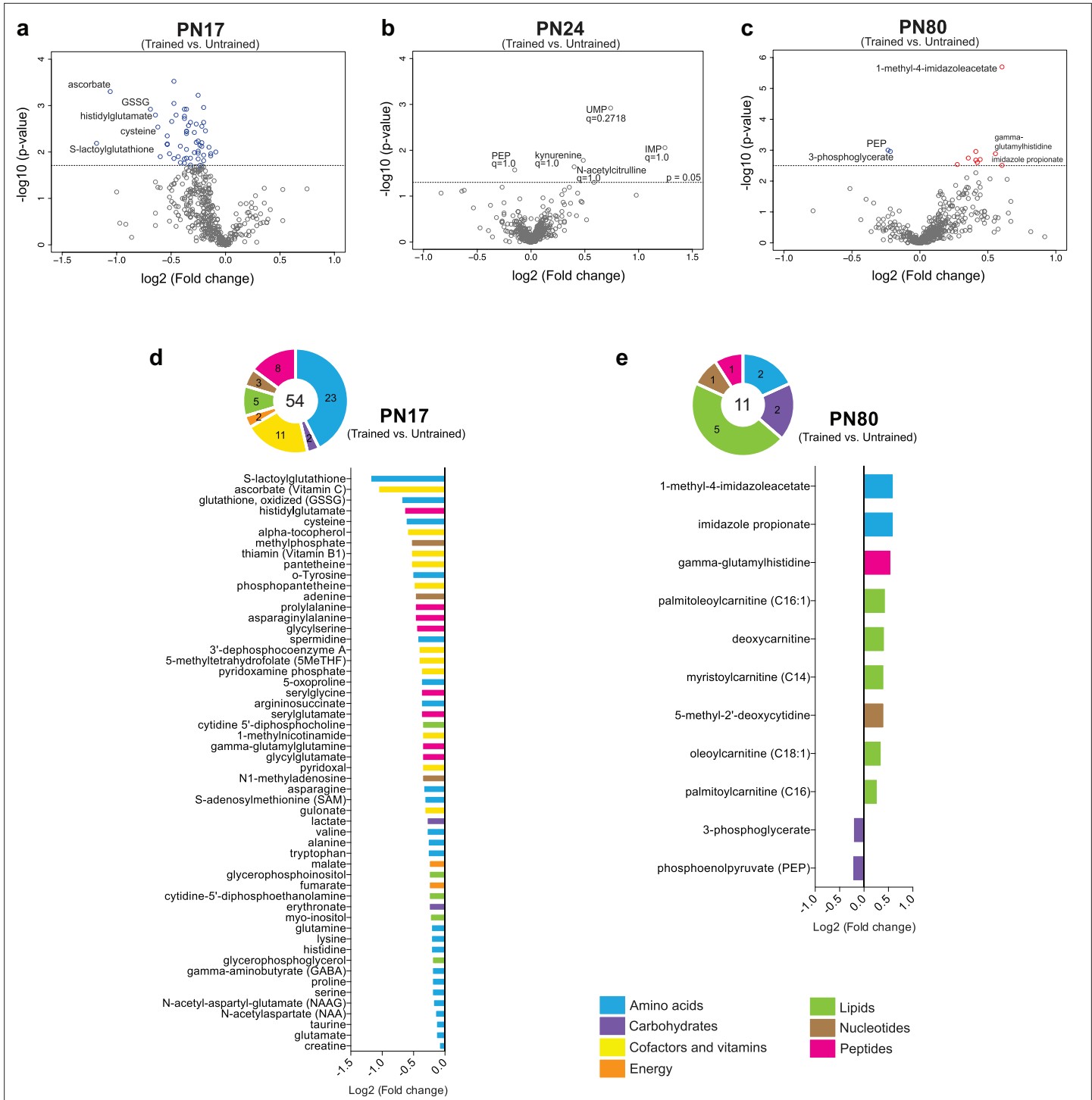

**Figure 5.** Metabolomic changes induced by learning in the hippocampus at postnatal day 17 (PN17), PN24, and PN80. (**a–c**) Volcano plots depicting the relationship between the fold changes and p-values between trained and untrained rats at PN17 (**a**), PN24 (**b**), and PN80 (**c**). In the volcano plots, each dot represents a specific metabolite. The y-axis shows the negative $\log_{10}$ of p-values (a higher value indicates greater significance) and the x-axis shows the $\log_2$ of the fold changes between values of mean peak intensities of trained rats relative to untrained rats. The dashed lines depict the thresholds of significance. Gray dots represent non-significant fold changes; blue and red dots represent significantly downregulated and upregulated metabolites, respectively. The volcano plots show the names of the top-5 downregulated or upregulated metabolites at the three different developmental ages. (**d, e**) The pie charts depict the numbers of metabolites significantly altered at PN17 (**d**) and PN80 (**e**), in the main endogenous pathways: amino acids, carbohydrates, cofactors and vitamins, energy-related metabolites, lipids, nucleotides, and peptides. The bar graphs depict the metabolites significantly altered 1 hr after learning at PN17 (**d**) and PN80 (**e**), in descending order of magnitude based on the $\log_2$(fold changes) between values of mean peak

*Figure 5 continued on next page*

*Figure 5 continued*

intensities of trained rats relative to untrained rats. Statistical significance: p < 0.05; q-value < 0.1 (two-tailed unpaired Welch's t-test and estimation of the false discovery rate [FDR]). n = 7 rats per group; one experiment. See *Figure 5—source data 1* for numerical values and detailed statistical information.

The online version of this article includes the following figure supplement(s) for figure 5:

**Source data 1.** Numerical values and detailed statistical information.

imidazole propionate 1 hr after training may be due to the high level of glutamate synthesis and utilization 1 hr after learning in adulthood. The increase in gamma-glutamylhistidine, a dipeptide composed of gamma-glutamate and histidine, after training at PN80 may reflect learning-induced incomplete breakdown product of protein degradation or learning-dependent synthesis of this peptide by formal condensation of the gamma-carboxy group of glutamate with the amino group of histidine (*Kakimoto and Konishi, 1976*). Furthermore, the learning-induced increase in the adult hippocampus of acyl-carnitines (i.e. palmitoleoylcarnitine [C16:1], myristoylcarnitine [C14], oleoylcarnitine [C18:1], and palmitoylcarnitine [C16]) suggests that training leads to an increase in beta-oxidation of fatty acids in hippocampal mitochondria (*Jones et al., 2010*). Catabolism of fatty acids could serve as an additional energy supply in adult brain after learning, but it remains unclear whether this increase is triggered by higher energy demand in trained animals or an adaptation to accept also lipids as energy substrates.

Finally, the significant decrease in two glycolytic metabolites, 3-phosphoglycerate and phosphoenol pyruvate, with adult learning (*Figure 5e*) suggests that glycolysis is differentially recruited in adult learning, in agreement with previous studies showing that glycolysis plays an essential role in brain plasticity and memory formation (*Suzuki et al., 2011*; *Newman et al., 2011*; *Descalzi et al., 2019*).

Future investigation shall analyze the relationship between developmental and learning-induced changes. These analyses require multiple comparisons at multiple ages. It could be important to know if changes associated with development and learning recruit similar or different mechanisms. Given our previous studies showing that biological and functional development of the hippocampus does not progress by default, but it is instructed by learning experiences, especially in infancy (*Bessières et al., 2020*), we suggest that several changes that occur over development will be also found to be associated with learning.

## Glutathione hippocampal metabolism is reduced in the infants relative to juveniles and adults

Among the 20 top-ranked amino acids, cofactors, and vitamins downregulated 1 hr after learning at PN17 (*Figure 5d*), a large proportion were molecules involved in the cellular defense against oxidative stress, and in particular metabolites belonging to the glutathione pathway. Hence, we focused on testing the regulation and functional role of this pathway in memory formation.

GSH is the most abundant antioxidant in the mammalian brain (*Dringen, 2000*). GSH plays a key role in protecting nerve cells against oxidative stress triggered by overproduction of ROS and harmful xenobiotics (*Dringen, 2000*; *Dringen and Hirrlinger, 2003*). As a redox regulator, GSH is a key modulator of the balance between cell proliferation and differentiation in dividing glial precursor cells, and it also regulates various signal transduction pathways and gene expression (*Dringen, 2000*; *Dringen and Hirrlinger, 2003*). Furthermore, through intra- and/or extracellular redox sites, GSH modulates the function of several ion channels and receptors implicated in neurotransmission, including the NMDA receptors (*Köhr et al., 1994*; *Janáky et al., 1999*; *Steullet et al., 2006*).

Although various brain cell types have different intrinsic glutathione-mediated antioxidant defense capacities, and are therefore differentially sensitive to oxidative stress, they are all equipped to biosynthesize and regenerate GSH from its oxidized form, GSSG (*Fernandez-Fernandez et al., 2012*; *Bolaños, 2016*). GSH is synthesized in a two-step pathway using glutamate, cysteine, and glycine as precursor amino acids. Glutamate and cysteine are combined via the action of the glutamate-cysteine ligase (GCL), a rate-limiting enzyme in GSH biosynthesis and composed of a catalytic subunit (GCLC) and a regulatory subunit (GCLM). The resultant dipeptide is then combined with glycine via a reaction mediated by glutathione synthetase (GSy) (*Figure 6a*). GSH is oxidized into GSSG by the glutathione peroxidase (GPx), a family of isoenzymes in which the GPx isoenzyme 1 is the most abundantly expressed isoform. GPx allows the detoxification of ROS such as hydrogen peroxide ($H_2O_2$),

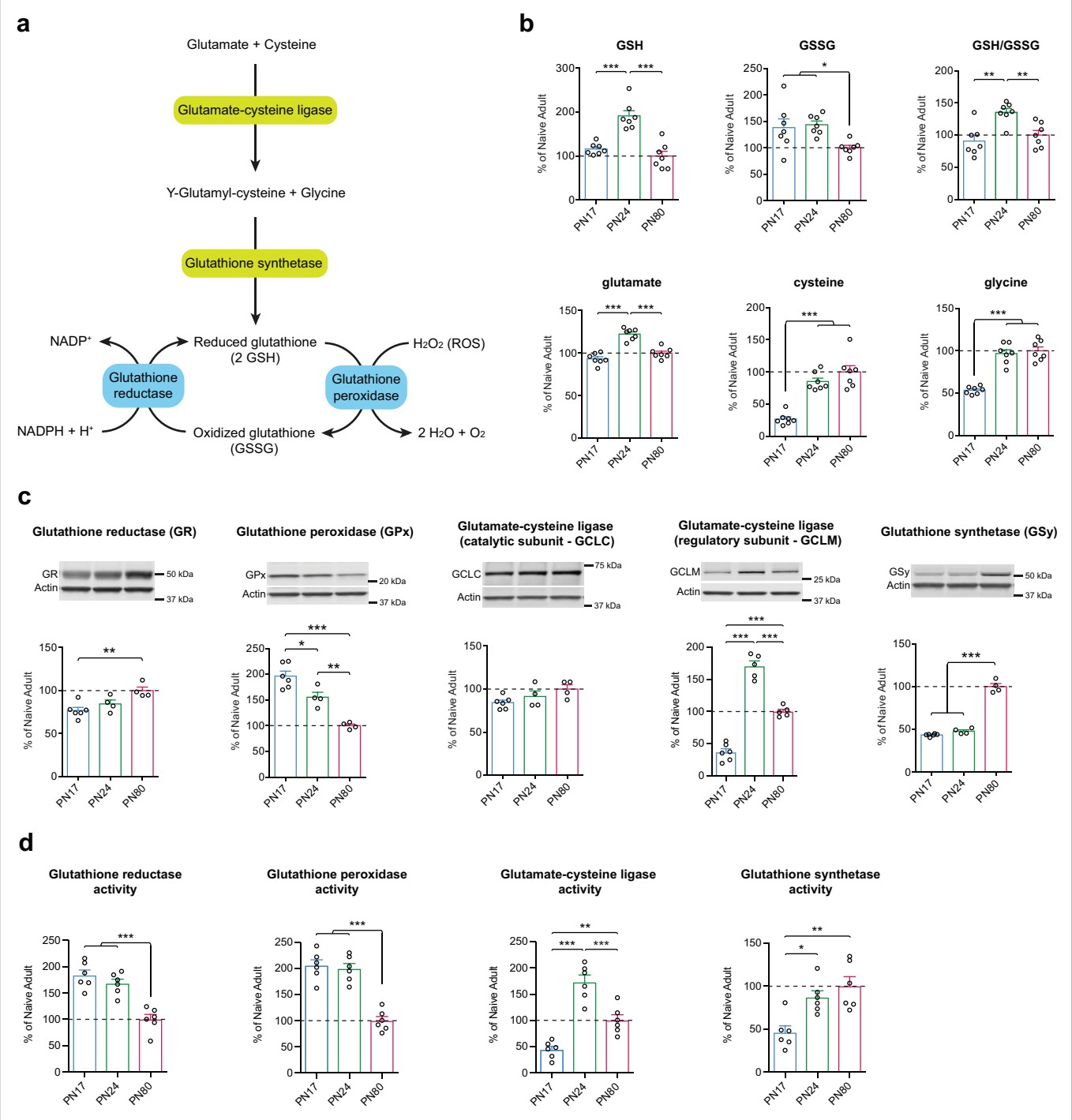

**Figure 6.** Glutathione-mediated antioxidant capacity is lower in the infant than in the juvenile and adult hippocampus. (**a**) Scheme depicting the glutathione oxidation/reduction cycle and biosynthesis pathway. (**b**) Quantification by ultrahigh performance liquid chromatography-tandem mass spectroscopy (UPLC-MS/MS) of reduced (GSH) and oxidized (GSSG) forms of glutathione, of the GSH/GSSG ratio, and of the GSH amino acid precursors glutamate, cysteine, and glycine in hippocampal samples from postnatal day 17 (PN17), PN24, and PN80 (n = 7 rats per group; one experiment). Data are expressed as mean percentage ± s.e.m. of the value in naive adult group (PN80). *p < 0.05; **p < 0.01, ***p < 0.001 (one-way ANOVA followed by Tukey's multiple comparisons test). (**c**) Representative examples and densitometric Western blot analyses of glutathione reductase (GR), glutathione peroxidase (GPx – isozyme 1), glutamate-cysteine ligase (catalytic subunit GCLC, and regulatory subunit GCLM), and glutathione synthetase (GSy), carried out with whole-protein extracts of dHC from PN17 (n = 6), PN24 (n = 4–5), and PN80 (n = 4–5) naive rats; two independent experiments. Actin was used as loading control. Data are expressed as mean percentage ± s.e.m. of the value in naive adult group. *p < 0.05, **p < 0.01, ***p < 0.001 (one-way ANOVA followed by Tukey's multiple comparisons test). (**d**) Mean activity of GR, GPx, GCL, and GSy (expressed as the

*Figure 6 continued on next page*

*Figure 6 continued*

quantity, in nmol, of substrate transformed per minute and normalized to the total quantity of proteins in each sample, in mg) assayed in dHC protein extracts from PN17 (n = 6), PN24 (n = 6), and PN80 (n = 6) naive rats; two independent experiments. Data are expressed as mean percentage ± s.e.m. of the value in naive adult group. *p < 0.05, **p < 0.01, ***p < 0.001 (one-way ANOVA followed by Tukey's multiple comparisons test). See *Figure 6— source data 1* and *Figure 6—source data 2* for numerical values and detailed statistical information.

The online version of this article includes the following source data for figure 6:

**Source data 1.** Numerical values.

**Source data 2.** Detailed statistical information.

into $H_2O$ and $O_2$. GSH is then regenerated by glutathione reductase (GR), using the pentose phosphate pathway product nicotinamide adenine dinucleotide phosphate (NADPH) as an electron donor (*Dringen and Hirrlinger, 2003*; *Fernandez-Fernandez et al., 2012*; *Bolaños, 2016*; *Figure 6a*).

To determine how glutathione metabolism changes over the course of the postnatal development, we first compared the hippocampal metabolomic levels and ratio of GSH and GSSG, along with the GSH precursor amino acids glutamate, cysteine, and glycine, at PN17, PN24, and PN80. As shown in *Figure 6b*, GSH level was significantly higher at PN24 compared to PN17 and PN80, whereas GSSG levels of PN17 and PN24 were significantly higher relative to that of PN80. As a result, the GSH/ GSSG mean ratio emerged as significantly higher at PN24 compared to PN80, and slightly but not significantly lower at PN17 relative to PN80. Despite the lack of significant decrease in the mean ratio of PN17 over PN80, there was a broader distribution of GSSG individual values at PN17 compared to PN80, suggesting that the regulation of the glutathione cycle might be different in the infant hippocampus. Furthermore, aside from the glutamate level, which reached a peak at PN24, the levels of the precursor amino acids cysteine and glycine were significantly lower at PN17 than at PN24 and PN80 (*Figure 6b*), suggesting either a lower de novo GSH synthesis capacity or a higher rate of consumption at PN17 relative to the juvenile and adult ages.

Western blot analysis of dorsal hippocampal extracts from PN17, PN24, and PN80 rats revealed that the level of GR was significantly lower at PN17 than in adulthood, whereas that of GPx isozyme 1 was 45% higher at PN17 compared to PN24 and almost 100% higher relative to PN80 (*Figure 6c*). These data suggested that there is a higher rate of glutathione oxidation in the infant brain. In addition, the levels of GCLC were similar across developmental ages, whereas the levels of GCLM at PN24 were 135% higher compared to PN17 and 70% higher compared to PN80. Finally, GSy was present at a higher concentration in adulthood than at infant and juvenile ages (*Figure 6c*). These results support the idea that the infant brain has a lower capacity for de novo GSH synthesis compared to juvenile and adult brains.

Finally, we determined how the enzymatic activity of GR, GPx, GCL, and GSy change across development in the dHC. The levels of GR and GPx activity were significantly higher at PN17 and PN24 compared to PN80 (*Figure 6d*), suggesting a higher rate of glutathione oxidation/reduction cycle during development compared to adulthood. Moreover, the level of activity of GCL dramatically increased between PN17 and PN24 and then decreased in adulthood at a level that however remained significantly elevated compared to PN17 (*Figure 6d*). Likewise, the activity of GSy was significantly higher at PN24 and PN80 compared to PN17. These results, in line with the protein expression levels (*Figure 6c*), suggest a reduced de novo GSH synthesis in infancy compared to juveniles and adults, which could be explained by a lower level of the rate-limiting GSH precursor cysteine and/or of GCLM at PN17 (*Figure 6b and c*).

Collectively, these results suggest that the infant hippocampus has a lower glutathione-mediated antioxidant defense capacity relative to the juvenile and adult hippocampi. Our data and conclusions are in agreement with previous studies showing that the developing brain is particularly vulnerable to free radicals, hence more susceptible to oxidative stress than later in life (*Buonocore et al., 2001*; *Khan and Black, 2003*; *Ikonomidou and Kaindl, 2011*; *Bakhtyukov et al., 2016*; *Galkina et al., 2017*).

## Infant learning increases neuronal GR activity

Metabolomic analysis showed that IA learning at PN17, but not at PN24 or PN80, significantly decreased the hippocampal level of GSSG without affecting the level of GSH, leading to a significant increase in the GSH/GSSG ratio 1 hr after infantile learning (*Figure 7a*).

Western blot analyses of dorsal hippocampal extracts extended these data revealing that the levels of GR, GPx isozyme 1, GCLC, GCLM, and GSy did not change either at 1 or 24 hr following IA learning at PN17 (*Figure 7b*). Hence, we hypothesized that the learning-induced increase in the GSH/GSSG ratio could result from changes in the enzymatic activity of GR, GPx, GCL, and/or GSy. Assessments at 15 min, 1 hr, 24 hr, or 7 days after IA training at PN17 revealed that only the activity of GR significantly increased at 15 min after training relative to untrained controls (*Figure 7c*) and remained significantly elevated at 1 and 24 hr after training before returning to baseline by 7 days post-training (*Figure 7c*). These results suggest that learning at PN17 induces an increase in the regeneration of GSH from GSSG, while it does not change the conversion of GSH into GSSG and the biosynthesis of GSH. The mechanisms by which GR activity increases following learning without a change in GR expression level itself are not known. We speculate that this increased activity may result from posttranslational modifications and/or allosteric interactions with the enzyme (*Deponte, 2013*; *Krauth-Siegel et al., 2005*). Despite GR exhibits several phosphorylation, acetylation, and ubiquitylation sites (*Deponte, 2013*), very little is known about the relevance of these posttranslational modifications in modulating GR activity, and the effects of learning on GR posttranslational modifications or allosteric modulations remain to be understood.

To determine whether the increase in GR activity was the result of an associative infantile learning as opposed to other footshock-induced activations, we repeated the experiment testing the changes in GR activity at 15 min post-training but included an additional control group of rats that underwent an immediate footshock experience (shock-only). In this protocol, the rats were exposed to a footshock immediately after being placed onto a grid, without allowing sufficient time to process the context-shock association (*Bessières et al., 2020*). No significant change relative to naive controls was observed in the shock-only control group (*Figure 7d*), leading to the conclusion that the increase in GR activity level was induced in response to an associative learning.

To determine whether the long-lasting increase in GR activity level following training is selective for early development and limited to the critical period of infantile amnesia, we conducted a similar experiment at PN24 as well as PN80, when rats are capable of expressing long-term memory. No changes were found in GR activity of rats trained at PN24 (*Figure 7e*) or PN80 (*Figure 7f*) indicating, in line with our metabolomic findings, that glutathione metabolism is differentially regulated in the rat hippocampus at PN17 relative to later ages.

As our biochemical analyses were obtained from hippocampal extracts, they did not dissect whether the learning-induced metabolomic changes and increase in GR activity are distinctively regulated within the different hippocampal subregions or cell types. Because the hippocampus comprehends several functional subregions and many cell types, it would be important that future studies investigate these specific subcomponent contributions. Our comprehensive hippocampal metabolomic results should provide important guidance and serve as a data platform for future analyses of subregion and cell type-specific metabolic pathways.

However, to begin investigating in which cell types infant learning induces metabolic changes, we isolated hippocampal astrocytes and neurons and determined whether the learning-induced increase in GR activity is differentially regulated in these cells. Dissecting the neuronal vs. astrocytic GR activity regulation following learning is important because previous studies have shown that despite the existence of several activity-dependent cell-autonomous mechanisms to maintain their redox balance (*Papadia et al., 2008*; *Bell and Hardingham, 2011*; *Baxter et al., 2015*), neurons have a limited intrinsic glutathione-dependent antioxidant capacity and their survival under high metabolic energy demands relies on the antioxidant protection promoted by surrounding astrocytes (*Herrero-Mendez et al., 2009*; *Fernandez-Fernandez et al., 2012*; *Bolaños, 2016*). In fact, astrocytes express significantly higher levels of various antioxidant molecules and ROS-detoxifying enzymes, including GSH, GCL, GPx, glutathione S-transferase, catalase, and thioredoxin reductase (*Bélanger et al., 2011*) in addition to the master antioxidant transcriptional activator nuclear factor-erythroid 2-related factor-2 (Nrf2) (*Fernandez-Fernandez et al., 2012*; *Jimenez-Blasco et al., 2015*). NMDAR-dependent Nrf2 activation in astrocytes stimulates the astrocytic release of GSH precursors, which neurons can use for

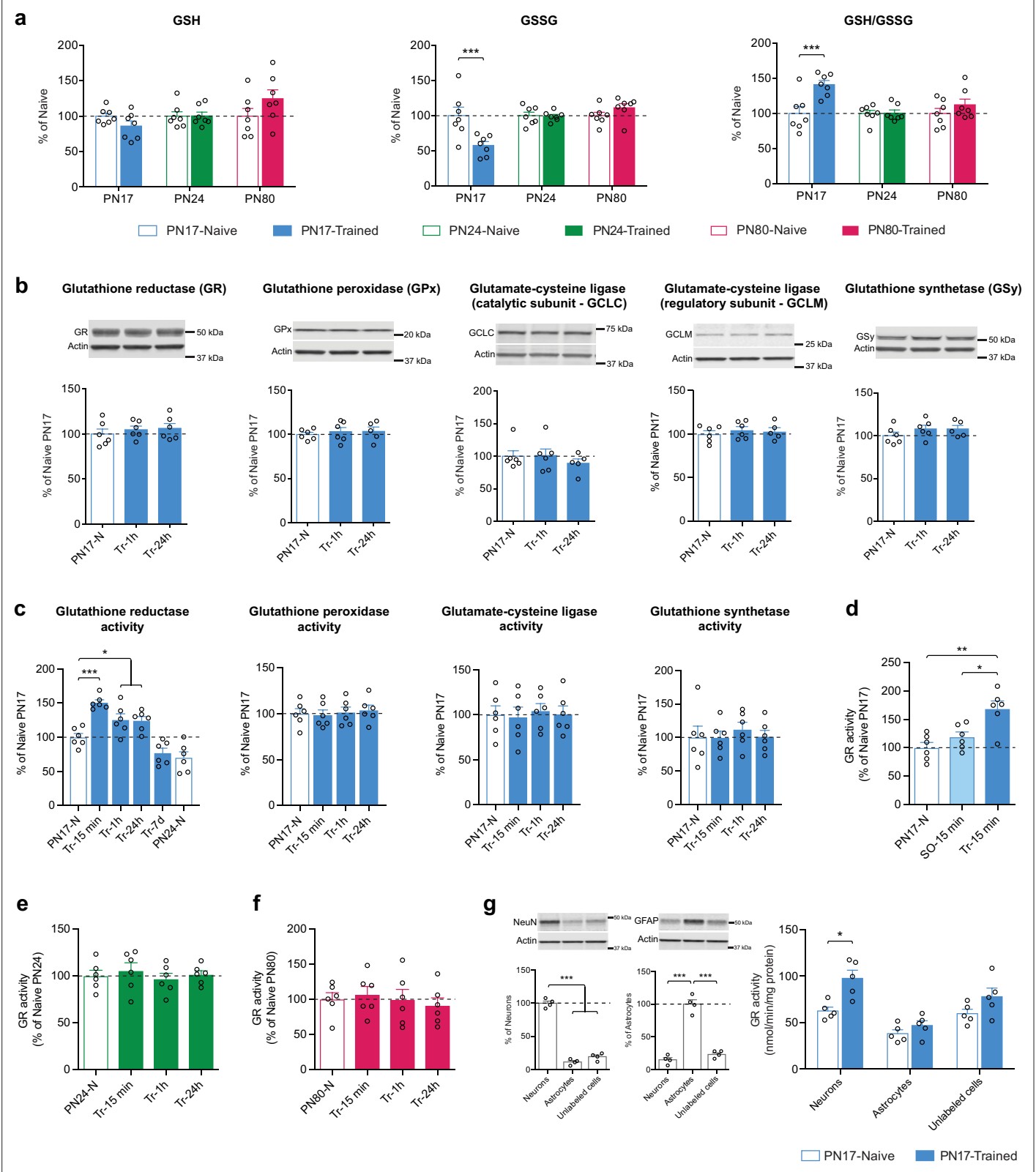

**Figure 7.** Infant episodic learning increases neuronal glutathione reductase (GR) activity. (**a**) Quantification by ultrahigh performance liquid chromatography-tandem mass spectroscopy (UPLC-MS/MS) of reduced (GSH) and oxidized (GSSG) forms of glutathione and the GSH/GSSG ratio in whole hippocampus from PN17, PN24, and PN80, in the untrained condition and 1 hr after inhibitory avoidance (IA) training (n = 7 rats per group). Data are expressed as mean percentage ± s.e.m. of the value in naive group at each age. ***p < 0.001 (two-way ANOVA followed by Bonferroni's multiple

*Figure 7 continued on next page*

*Figure 7 continued*

comparisons test). (**b**) Representative examples and densitometric Western blot analyses of GR, glutathione peroxidase isozyme 1 (GPx), glutamate-cysteine ligase (catalytic subunit GCLC, and regulatory subunit GCLM), and glutathione synthetase (GSy), carried out with whole-protein extracts of dHC from postnatal day 17 naive rats (PN17-N) and rats trained in IA at PN17 and euthanized 1 hr (Tr-1h) or 24 hr (Tr-24h) after training (n = 5–6 rats per group; two independent experiments). Actin was used as the loading control. Data are expressed as mean percentage ± s.e.m. of the value in PN17 naive rats. p > 0.05 (one-way ANOVA followed by Tukey's multiple comparisons test). (**c**) Mean activity of GR, GPx, GCL, and GSy (expressed as the quantity, in nmol, of substrate transformed per minute and normalized to the total quantity of proteins in each sample, in mg) assayed in dHC protein extracts obtained from rats trained in IA at PN17 and euthanized 15 min, 1 hr, 24 hr, or 7 days after training (Tr-15min, Tr-1h, Tr-24h, and Tr-7d, respectively) (n = 6 rats per group; two independent experiments). To account for developmental differences, two groups of naive (N) rats were used: PN17 and PN24 (n = 6 rats per group). Data are expressed as mean percentage ± s.e.m of the value in PN17 naive rats. *p < 0.05, ***p < 0.001: significance vs. PN17 naive rats (one-way ANOVA followed by Dunnett's multiple comparisons test); p > 0.05 for the comparison between PN24-N and Tr-7d groups (two-tailed unpaired Student's t-test). (**d**) GR activity carried out with dHC protein extracts obtained from PN17 naive rats (PN17-N), from rats exposed to an immediate footshock without IA-context exposure (shock-only) and euthanized 15 min later (SO-15min) or from rats trained in IA at PN17 and euthanized 15 min later (Tr-15min) (n = 6 rats per group; two independent experiments). GR activity is expressed as mean percentage ± s.e.m of the value in PN17 naive rats. *p < 0.05, **p < 0.01 (one-way ANOVA followed by Tukey's multiple comparisons test). (**e**) GR activity assayed in dHC protein extracts obtained from PN24 naive rats (PN24-N) and rats trained in IA at PN24 and euthanized 15 min, 1 hr, or 24 hr after training (Tr-15min, Tr-1h, and Tr-24h, respectively) (n = 6 rats per group; two independent experiments). GR activity is expressed as mean percentage ± s.e.m of the value in PN24 naive rats. p > 0.05 (one-way ANOVA followed by Dunnett's multiple comparisons test). (**f**) GR activity assayed in dHC protein extracts obtained from PN80 naive rats (PN80-N) and rats trained in IA at PN80 and euthanized 15 min, 1 hr, or 24 hr after training (Tr-15min, Tr-1h, and Tr-24h, respectively) (n = 6 rats per group; two independent experiments). GR activity is expressed as mean percentage ± s.e.m of the value in PN80 naive rats. p > 0.05 (one-way ANOVA followed by Dunnett's multiple comparisons test). (**g**) Representative examples and densitometric Western blot analyses of NeuN and GFAP carried out with whole-protein extracts of fluorescence-activated cell sorting (FACS)-sorted neurons (NeuN+), astrocytes (GFAP+), and unlabeled counter-selected cells from the dHC of PN17 naive rats (n = 4 x 3 rats per group; two independent experiments). Actin was used as the loading control. NeuN intensity values are expressed as mean percentage ± s.e.m. of the value in neurons (NeuN+) group; GFAP intensity values are expressed as mean percentage ± s.e.m. of the value in astrocytes (GFAP+) group. ***p < 0.001 (one-way ANOVA followed by Tukey's multiple comparisons test). GR activity carried out with whole-protein extracts of FACS-sorted neurons (NeuN+), astrocytes (GFAP+), and unlabeled cells from the dHC of PN17 naive rats (PN17-N) and of rats trained in IA at PN17 and euthanized 15 min later (Tr-15min) (n = 5 x 3 rats per group; two independent experiments). GR activity is expressed as mean nmol/min/mg protein± s.e.m. *p < 0.05 (two-way ANOVA followed by Bonferroni's multiple comparisons test). See ***Figure 7—source data 1***, ***Figure 7—source data 2*** for numerical values and detailed statistical information.

The online version of this article includes the following figure supplement(s) for figure 7:

**Source data 1.** Numerical values.

**Source data 2.** Detailed statistical information.

de novo GSH biosynthesis (***Dringen et al., 1999***; ***Jimenez-Blasco et al., 2015***). Moreover, compared to neurons, astrocytes rely more on glycolysis, rather than oxidative phosphorylations, for energy generation (***Herrero-Mendez et al., 2009***) and have a mitochondrial respiratory chain that is associated with a higher production of mitochondrial ROS (***Lopez-Fabuel et al., 2016***). Astrocytic ROS regulate glucose utilization, boost the non-cell-autonomous Nrf2-driven antioxidant support to neurons, and are required for memory formation in mouse (***Vicente-Gutierrez et al., 2019***). Yet, a possible role of astrocytes in the antioxidant defense during infantile memory formation remains to be determined.

We measured GR activity in fluorescence-activated cell sorting (FACS)-sorted neurons, astrocytes, and unlabeled counter-selected brain cell types (other cells, including oligodendrocytes, microglia, vascular cells) from the dHC of untrained and trained PN17 rats euthanized 15 min after training, a time point at which the induction of GR activity peaks in total dHC extracts. We found that infant learning induces a significant increase in GR activity in neurons but not in astrocytes or other cells (***Figure 7g***). Notably, the basal level of GR activity appears to be higher in neurons than in astrocytes despite the absence of statistical significance (***Figure 7g***). These results are in line with previous studies showing that the expression of GR is higher in neurons compared to astrocytes (***Gutterer et al., 1999***) and that neurons compensate their low level of GCL and GSH concentration by a very efficient GSH regenerative system (***Flohé et al., 2011***; ***Rodriguez-Rodriguez and Almeida, 2013***).

## Learning-induced increase in GR activity is required for long-term memory formation in infant rats

Finally, we investigated whether the learning-induced increase in GR activity is differentially required for the formation of infantile memory. Toward this end, rats received a bilateral dorsal hippocampal injection of the selective, cell-permeable GR inhibitor 2-acetylamino-3-[4-(2-acetylamino-2-carboxy

ethylsulfanylthiocarbonylamino)phenylthio-carbamoylsulfanyl]-propionic acid (2-AAPA) at either 100 or 200 µM 15 min before IA training at PN17. While 100 µM prevented only the learning-dependent increase in GR activity, 200 µM decreased GR activity by more than 75% (*Figure 8a*). Both doses completely blocked memory formation in rats trained at PN17 (*Figure 8b*). Infantile memory formation was determined as previously established (*Travaglia et al., 2016b*; *Bessières et al., 2020*), that is, by memory reinstatement given 7 days after IA training (presentation of context [T1] followed 2 days later, by a reminder shock – RS), a protocol that significantly reinstates the apparently forgotten memory in rats injected with vehicle (T2, *Figure 8b*). The 2-AAPA-injected PN17 rats re-learned the IA task when retrained (Tr) upon entering the shock compartment at T2, demonstrating that the 2-AAPA injections did not disrupt hippocampal function (T3, *Figure 8b*). Conversely, bilateral injections of 2-AAPA (100 µM) in the dHC of rats trained at PN24 (*Figure 8c*) or at PN80 (*Figure 8d*) did not prevent memory formation. We concluded that the learning-induced increase in GR activity is necessary to form IA memory selectively in infant rats but is no longer required in juveniles and adults.

Furthermore, to dissect how infant learning and GR inhibition affect the levels of GSH and GSSG, PN17 rats were injected bilaterally in the dHC with vehicle or 2-AAPA (100 µM) 15 min before IA learning and a fluorometric assay was carried out on their dHC extracts collected 15 min, 1 hr, or 24 hr following training (*Figure 8e–g*). We found that infant learning significantly increases GSH level (*Figure 8e*) while decreasing GSSG (*Figure 8f*) at 15 min after learning. While the level of GSH was back to basal level at 1 and 24 hr after training (*Figure 8e*), at the same time points, the levels of GSSG remained significantly lower than those of untrained control rats (*Figure 8f*). This time course is consistent with our metabolomic data showing that GSSG was significantly decreased while GSH was unchanged relative to untrained controls at 1 hr after training (*Figure 7a*). However, the reason why GSH rapidly returned to basal level (at the 1 hr time point) while GSSG remains reduced and GR activity elevated compared to untrained rats remains unknown. One explanation could be that GSH is rapidly consumed between 15 min and 1 hr to maintain the redox balance or to respond to additional unknown metabolic requirements following learning. These learning-induced changes in GSH and GSSG were prevented by 2-AAPA-mediated GR inhibition (*Figure 8e–g*), providing evidence that the learning-induced GSH increase and GSSG decrease were mediated by an increase in GR activity following learning.

Consistent with these results, previous studies have reported that a transient GSH deficit during critical periods of neurodevelopment significantly affects several brain maturation processes, leading to immediate and delayed negative consequences on cognitive functions, including impairment of working memory, as well as spatial and episodic memory formation during infancy, adolescence, and adulthood (*Cabungcal et al., 2007*; *Kulak et al., 2013*). Notably, memory impairments induced by reduction in the antioxidant capacity of the GSH system can be partly compensated by de novo synthesis of ascorbic acid (*Dalton et al., 2000*). In our study, inhibition of GR during infantile learning resulted in memory impairment, revealing that the ascorbic acid was not sufficient to compensate for the decrease in GSH regeneration capacity. In fact, ascorbic acid was also significantly downregulated after training at PN17 (*Figure 5a and d*).

The observation that the transient inactivation of the GR affects memory formation in infant but not in juvenile or adult rats strongly suggests that the hippocampal memory system is more vulnerable to redox dysregulation during infancy than at later ages. This result may reflect a lower antioxidant defense capacity in infancy relative to later in life, in the context of an already high baseline level of metabolic activity. The higher vulnerability of the infant brain to oxidative stress may also be reflective of the immaturity of the astrocytic-neuronal shuttling of GSH precursors (*Bélanger et al., 2011*) leading to less antioxidant support supplied by astrocytes, which are still proliferating during this period (*Bandeira et al., 2009*). Indeed, both the activity-dependent intrinsic neuronal defenses, in combination with the non-cell-autonomous Nrf2-driven antioxidant support from astrocytes, have been shown to be optimal for maintaining neuronal redox balance in the face of oxidative stress (*Fernandez-Fernandez et al., 2012*; *Baxter et al., 2015*).

Finally, our results showing that inhibiting GR at the time of learning in juvenile or adult rats has no effect on memory performances could be explained by a higher level of GCL and GSy activities (as shown *Figure 6*) and/or of differential antioxidant mechanisms present in the adult and juvenile hippocampi. Future investigations will determine whether learning in juveniles and adults affects the activity of GCL and GSy and if these activities are required for memory formation. In a previous study,

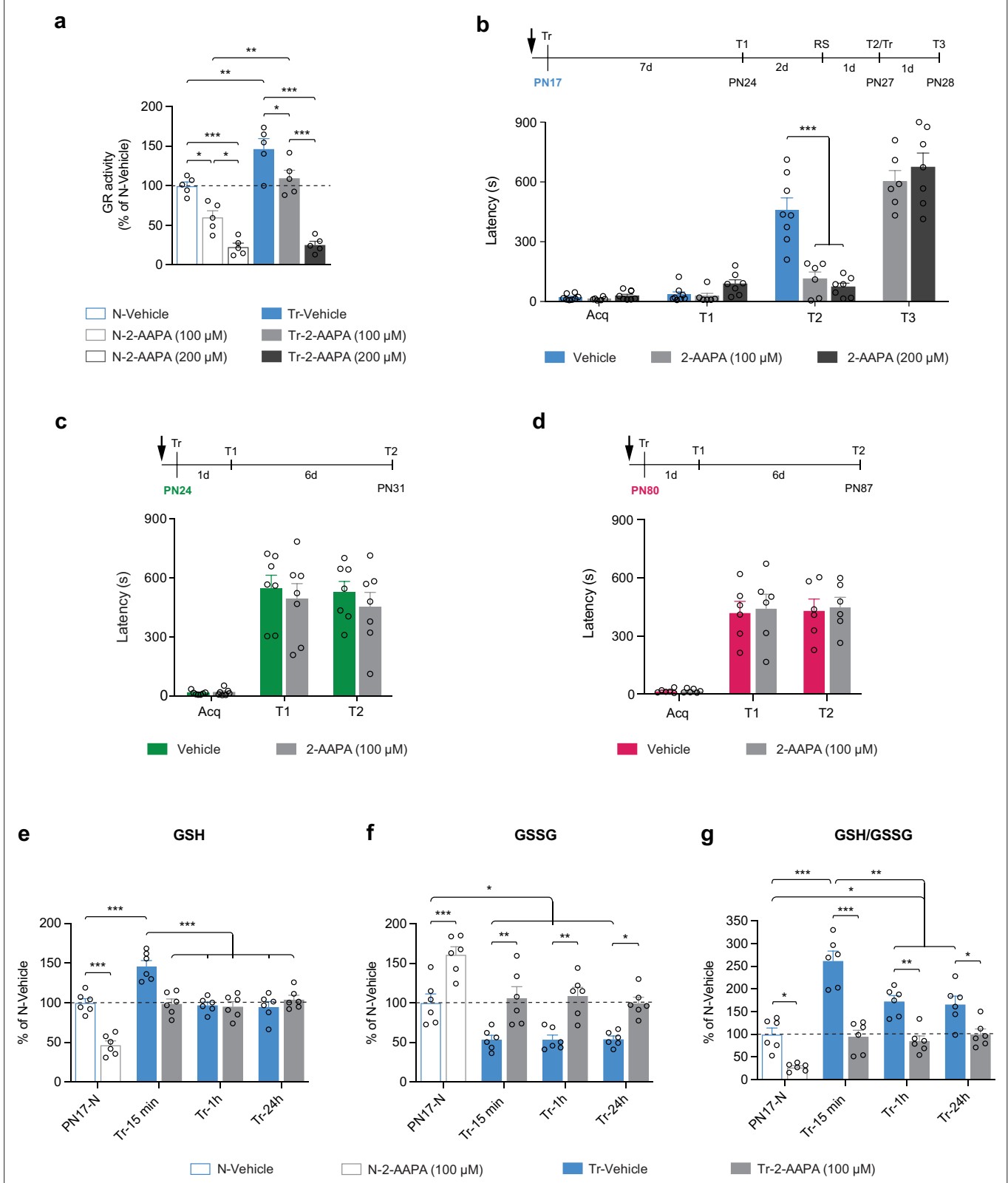

**Figure 8.** Learning-induced increase in glutathione reductase (GR) activity is required for long-term memory formation in infant rats. (**a**) GR activity carried out with dorsal hippocampus (dHC) protein extracts obtained from PN17 naive (N) and trained (Tr) rats that received a bilateral hippocampal injection of the selective GR inhibitor 2-acetylamino-3-[4-(2-acetylamino-2-carboxyethylsulfanylthiocarbonylamino)phenylthio-carbamoylsulfanyl]-propionic acid (2-AAPA) (100 or 200 µM) or Vehicle 15 min before inhibitory avoidance (IA) training. Rats were euthanized 15 min after training (n = 5 rats

*Figure 8 continued on next page*

*Figure 8 continued*

per group; two independent experiments). GR activity is expressed as mean percentage ± s.e.m. of the value in N-Vehicle group. *p < 0.05, **p < 0.01, ***p < 0.001 (one-way ANOVA followed by Tukey's multiple comparisons test). (**b**) Mean latency ± s.e.m. (in seconds, s) of rats injected in the dHC (black arrow) with vehicle or 2-AAPA (100 or 200 μM) 15 min before IA training at PN17 (n = 6–8 rats per group; three independent experiments). Acq: latency to enter the dark side of the chamber at training (Tr). Rats were tested 7 days later (**T1**), and 2 days later received a reminder shock (RS), followed by another memory retention test 1 day later (**T2**). At T2, upon entering the shock compartment, rats were trained again (Tr) and tested 1 day later (**T3**). ***p < 0.001 (two-way repeated measures (RM) ANOVA followed by Bonferroni's multiple comparisons test). (**c–d**) Mean latency ± s.e.m. of rats injected in the dHC (black arrow) with vehicle or 2-AAPA (100 μM) 15 min before IA training at PN24 (**c**), n = 7 rats per group (two independent experiments); or at PN80 (**d**), n = 6 rats per group (two independent experiments). Memory was tested 1 day after training (**T1**) and again 6 days later (**T2**). p > 0.05 (two-way RM ANOVA followed by Bonferroni's multiple comparisons test). (**e–g**) Quantification of reduced form of glutathione (GSH) (**e**), oxidized form of glutathione (GSSG) (**f**) and of the GSH/GSSG ratio (**g**) in the dHC of PN17 naive rats (PN17-N) and rats trained in IA at PN17 and euthanized 15 min, 1 hr, or 24 hr after training (Tr-15min, Tr-1h, Tr-24h, respectively) (n = 6 rats per group; two independent experiments). Rats received a bilateral hippocampal injection of the selective GR inhibitor 2-AAPA (100 μM) or vehicle 15 min before IA training at PN17. Data are expressed as mean percentage ± s.e.m. of the value in N-Vehicle group. *p < 0.05, **p < 0.01, ***p < 0.001 (two-way ANOVA followed by Bonferroni's multiple comparisons test). See *Figure 8— source data 1*, *Figure 8—source data 2* for numerical values and detailed statistical information.

The online version of this article includes the following figure supplement(s) for figure 8:

**Source data 1.** Numerical values.

**Source data 2.** Detailed statistical information.

it has been shown that a moderate decrease of neuronal GSH in CA1 area of the hippocampus of adult mice by GCLC knockdown leads to dendrite disruption and cognitive impairment, revealing that hippocampal neurons of adult mouse require a large pool of glutathione to sustain cognitive functions (*Fernandez-Fernandez et al., 2018*).

## Conclusion

Identifying the metabolic changes of the brain over development is key for a better understanding of the biology underlying the maturation of brain functions. As many neurodevelopmental disorders seem to be linked to metabolic dysregulations, knowing the evolution of brain metabolomics over development should provide novel insights for studying these diseases. We approached this metabolomic investigation at the whole rat hippocampus level over five developmental ages, which provided a comprehensive view of the hippocampal metabolomic changes over developmental and with learning. Guided by our current data, future investigations shall dissect how development and learning affect the metabolomic profile of specific cell types in the different hippocampal subregions.

Here, we found a higher diversity and concentration of metabolites at juvenile age (PN24), which declined in adulthood. Most of the amino acids and nucleotides decrease, while the majority of lipids and cofactors/vitamins increase over the course of the hippocampus postnatal development. Our results also showed that learning at different ages differentially regulates the hippocampal metabolome. Significant metabolite changes following learning were observed at PN17 and PN80, mainly involving amino acid, cofactor and vitamin, peptide, and lipid metabolisms. Our metabolomic analyses revealed that learning in infancy changes the levels of many metabolites belonging to the glutathione pathway.

Additional molecular, biochemical, and behavioral investigations demonstrated that infant learning induces a fast and long-lasting increase in the activity of the neuronal GR in the dHC, and that this activity is critical for the formation of long-term memory in infant rats, whereas it becomes dispensable in juveniles and adults. These results indicate, for the first time to our knowledge, that GR, and hence the glutathione pathway, plays a differential role in long-term memory formation in the infant hippocampus.

Although the mechanisms by which learning-induced activation of GR supports long-term memory formation in healthy infant hippocampi remain to be understood, some speculations are possible. Cell types particularly sensitive to GSH regulation are the parvalbumin-immunoreactive GABAergic interneurons (PV cells), the oligodendrocytes, and their progenitor cells (*Do et al., 2009*; *Steullet et al., 2016*). Therefore, it has been suggested that a developmental GSH deficit would disrupt PV cells and cause anomalies in myelination, which would in turn interfere with normal brain maturation, resulting in the onset of learning disabilities and neurodevelopmental psychiatric disorders such as schizophrenia and bipolar disorders later in life (*Do et al., 2009*; *Kulak et al., 2013*; *Steullet et al., 2016*;

*Hardingham and Do, 2016*). In light of the crucial role played by PV cells and myelination in sensory systems to regulate critical periods for plasticity and acquisition of functional competences (*Takesian and Hensch, 2013*; *Do et al., 2015*), we speculate that the GSH-deficit mediating memory impairment in infancy could be due to the disruption of the initiation or closure of specific critical periods in the hippocampal memory system. This hypothesis is consistent with the idea that neurodevelopmental disorders are, to some extent, the result of defective critical periods (*Alberini and Travaglia, 2017*). In sum, our metabolomic profiling of the rat hippocampus indicated that the ensemble of metabolites changes greatly over development, and that distinct metabolic pathways, such as the glutathione metabolism, are differentially regulated and engaged by learning at different ages.

## Materials and methods

### Animals

All animal procedures complied with the US National Institute of Health Guide for the Care and Use of Laboratory Animals and were approved by the New York University Animal Care Committees. One-, 7-, 17-, and 24-day-old male and female rats were obtained from E9–E10 pregnant Long-Evans female rats (purchased from Charles River Laboratories). Pre-weaning rats were housed with their littermates and mother in 30.8 cm × 40.6 cm × 22.2 cm plastic cages containing ALPHA-dri bedding. The birth date was defined as PN0, and the litters were culled to 10–12 pups (6 males and 6 females, if applicable) on PN1. After weaning (PN21), rats were group-housed (two to five animals per cage). Adult male Long-Evans rats (Envigo) weighing 200–250 g were used as PN80 specimens. Adult rats were housed two per cage. All rats were maintained on a 12 hr light/dark cycle with ad libitum access to food and water. All experiments were carried out during the light cycle.

### Inhibitory avoidance

IA was carried out as previously described (*Travaglia et al., 2016a*) in an IA chamber (Med Associates Inc, St. Albans, VT) consisting of a rectangular Perspex box divided into a safe compartment and a shock compartment (each 20.3 cm × 15.9 cm × 21.3 cm). The safe compartment was white and illuminated, and the shock compartment was black and dark. The apparatus was located in a sound-attenuated room with dim red light illumination. Animals were handled once a day for 3–5 min for 3 (pups) or 5 (adults) days before any behavioral procedure. During training sessions, each rat was placed in the safe compartment with its head facing away from the door. After 10 s, the door separating the compartments was automatically opened, allowing the rat access to the shock compartment. The door closed automatically when the rat crossed with all four limbs the invisible infrared light photosensors located in the shock compartment. Two seconds later, a footshock (2 s, 1 mA) was automatically delivered to the grid floor of the shock chamber via a constant current scrambler circuit. The animal remained in the dark compartment for additional 10 s, and was then returned to its home cage until testing for memory retention or euthanasia at the designated time points. As controls, we used untrained animals (handled and then returned to their home cage, termed naive) and rats exposed to a footshock without the IA context experience (shock-only). Shock-only treatment consisted of placing the rat onto grid of the shock compartment and immediately delivering a footshock of the same duration and intensity used in IA training. The animal returned to its home cage immediately after the footshock delivery. This protocol does not induce any association between the context and the foot shock. Retention tests were performed by placing the rat back in the safe compartment and measuring the latency to enter the dark compartment. Footshocks were not administered during the retention tests, and testing was terminated at 900 s. During retraining sessions, rats tested for memory retention received a footshock upon entering the dark compartment. Locomotor activity was measured during training and testing by automatically counting the number of times each rat crossed the infrared light photosensors located in both the safe and the shock compartment. Reminder footshocks (RS), with identical duration and intensity to those used in training, were administered in a novel neutral chamber with transparent walls, located in a different white light illuminated experimental room. For the metabolomics and biochemical studies, rats were not tested for memory retention to avoid the confound of changes resulting from testing. All behavioral tests were carried out blind to training and/ or treatment conditions.

## Hippocampal extract preparation for metabolomic profiling

Rats were euthanized by decapitation, and their brains were quickly removed and placed in ice-cold phosphate-buffered saline (PBS 1×), pH 7.4. For each animal, the whole hippocampi (including both dorsal and ventral subregions) were dissected and immediately snap-frozen in isopentane on dry ice. All samples were stored at –80°C until processing. Frozen samples were shipped in dry ice to Metabolon Inc (Morrisville, NC) for metabolomic analysis using a proprietary methodology. Extraction of samples was performed using a MicroLab STAR automated liquid handling robot (Hamilton Robotics, Inc, Reno, NV); 450 µL of methanol was added to 100 µL of sample to precipitate proteins. Four recovery standards (DL-2-fluorophenylglycine, tridecanoic acid, cholesterol-d6 and 4-chlorophenylalanine) were added to each sample to determine extraction efficiency. To remove proteins, dissociate small molecules bound to protein, and recover metabolites, proteins were precipitated with methanol under vigorous shaking for 2 min (Glen Mills GenoGrinder 2000) followed by centrifugation (1300 *g* for 10 min at room temperature [RT]). The resultant supernatants were placed in a TurboVap (Zymark) to remove the organic solvent. Each sample extract was then divided into five equal aliquots (see the *Metabolomic profiling* section), dried under nitrogen, and stored *in vacuo* prior to metabolomic profiling.

## Metabolomic profiling

Metabolomic profiling was performed by untargeted UPLC-MS/MS, which detected eight main classes of metabolites: amino acids, carbohydrates, nucleotides, peptides, lipids, cofactors or vitamins, metabolites involved in energy metabolism, and xenobiotics. Detailed descriptions of the protocols were published previously (*Evans et al., 2009*; *Bridgewater, 2014*). Briefly, UPLC-MS/MS was performed on a Waters Acquity UPLC (Waters, Milford, MA) and a Thermo Fisher Scientific Q-Exactive high-resolution mass spectrometer (Waltham, MA) interfaced with a heated electrospray ionization (HESI-II) source and Orbitrap mass analyzer. Aliquots previously dried under nitrogen were reconstituted in solvents compatible with four different methods described below; each reconstitution solvent contained a series of standards at fixed concentrations to ensure injection and chromatographic consistency. The first two aliquots were analyzed by two separate reverse-phase (RP)/UPLC-MS/MS methods with positive-ion mode electrospray ionization (ESI): one aliquot was analyzed using acidic positive-ion conditions, chromatographically optimized for more hydrophilic compounds. In this method, the extract was gradient-eluted from a C18 column (2.1 mm × 100 mm Waters UPLC BEH C18 1.7 µm column held at 40°C) using 5% water, 95% methanol and containing 0.05% perfluoropentanoic acid (PFPA) and 0.1% formic acid (FA) at pH 3.5. The second aliquot was also analyzed using acidic positive-ion conditions; however, it was chromatographically optimized for more hydrophobic compounds. In this method, the extract was gradient-eluted from the aforementioned C18 column using methanol, acetonitrile, water, 0.05% PFPA, and 0.01% FA at pH 3.5. A third aliquot was analyzed by RP/UPLC-MS/MS using basic negative-ion mode ESI in a separate dedicated C18 column. The basic extracts were gradient-eluted from the column using methanol (95%) and water (5%) with 6.5 mM ammonium bicarbonate at pH 8. The fourth aliquot was analyzed via negative ionization following elution from a Hydrophilic-Interaction Chromatography (HILIC) column (2.1 mm × 150 mm Waters UPLC BEH Amide, 1.7 µm column held at 40°C) using a mobile phase consisting of 10 mM ammonium formate in 15% water, 5% methanol, 80% acetonitrile (pH 10.8). For the four aliquots previously mentioned, the sample injection volume was 5 µL using a 2× needle loop overfill. A fifth aliquot was reserved for backup. Several types of quality control (QC) samples were analyzed in concert with the experimental samples: a pooled matrix sample (PMS) generated by taking a small volume of each experimental sample served as a technical replicate throughout the dataset; extracted water samples served as process blanks; and a cocktail of QC standards that were carefully chosen not to interfere with the measurement of endogenous compounds were spiked into every analyzed sample, allowing monitoring of instrument performance and chromatographic alignment. Instrument variability was determined by calculating the median relative standard deviation (RSD) of peak area for each internal standard that was added to each sample prior to injection into the mass spectrometers (RSD: 3%). Overall process variability was determined by calculating the median RSD for all endogenous metabolites (i.e., non-instrument standards) present in 100% of the PMS (RSD: 8%). Both RSD values met Metabolon's acceptance criteria. Experimental samples were randomized across the platform run, with QC samples spaced evenly among the injections. The scan range varied slightly

between methods but covered a mass-to-charge ratio (m/z) of 70–1000 with a scan speed of ~9 scans per second (alternating between MS and MS/MS scans) and a mass resolution of 35,000 (measured at 200 m/z). Mass calibration was performed as needed to maintain <5 ppm mass error for all standards monitored.

## Metabolite identification and quantification

Data processing, including chromatographic alignment, QC utilization, and metabolite identification has been performed as previously described (*Evans et al., 2009*; *Bridgewater, 2014*; *Dehaven et al., 2010*). The samples (including QC samples) were chromatographically aligned based on a retention index (RI) that used internal standards assigned with a fixed RI value. Spectral peaks were identified using Metabolon's proprietary peak detection and integration software and were quantified using area-under-the-curve values. Criteria for peak detection included thresholds for signal-to-noise ratio, area, and width. Metabolites were identified by automated comparison of the ion features in the experimental samples to a reference library of purified chemical standard entries developed at Metabolon. Biochemical identification was based on the combination of three criteria: RI values, (m/z) ratio, and the MS/MS forward and reverse scores between the experimental data and the standards present in the library (*Dehaven et al., 2010*). The MS/MS scores were based on the comparison of the ions present in the experimental spectrum to the ions present in the library spectrum. All proposed identifications were then manually reviewed and hand-curated based on the criteria described above. For metabolites that were not covered by the standards, additional library entries were added based on their recurrent and unique chromatographic and mass spectral signatures. For studies spanning multiple days, a data normalization step was performed to correct for variation resulting from inter-day differences in instrument tuning. Essentially, each compound was corrected in run-day blocks by registering the medians to equal 1.00 and normalizing each data point proportionately. This normalization minimizes any inter-day instrument gain or drift, but does not interfere with intra-day sample variability.

## Hippocampal cannula implants and injections

On PN15, PN22, or PN80, rats were anesthetized with isoflurane mixed with oxygen. Stainless steel cannulas (26-gauge) were implanted bilaterally in the dHC (for PN15: –3.0 mm anterior, 2.2 mm lateral, and –2.3 mm ventral from bregma; for PN22: –3.4 mm anterior, 2.2 mm lateral, and –2.5 mm ventral from bregma; for PN80: –4.0 mm anterior, 2.5 mm lateral, and –2.5 mm ventral from bregma) through holes drilled in the overlying skull. The cannulas were fixed to the skull with dental cement. At the end of the surgery, rats were returned to the dam and littermates (PN15) or their home cage (PN22 and PN80) for a 2-day recovery period for the pups or a 10-day recovery period for the adults prior to further experimental manipulations. Hippocampal injections were performed using a 33-gauge needle that extended 1 mm beyond the tip of the guide cannula and was connected via polyethylene tubing to a Hamilton syringe. Injections were delivered using an infusion pump at a rate of 0.1 μL/min to deliver a total volume of 0.3 μL per side over 3 min. The injection needle was left in place for 2 min after injection to allow complete diffusion of the solution. The potent, selective, cell-permeable GR inhibitor 2-AAPA (Millipore-Sigma, cat# A4111) (*Seefeldt et al., 2009*; *Zhao et al., 2009*) was dissolved in PBS 1× containing 10% dimethylsulfoxide. On each side, 0.3 μL 2-AAPA was injected at 100, or 200 μM in independent groups of rats, 15 min before IA training. To verify proper placement of the cannula implants, rats were euthanized at the end of the behavioral experiments, and their brains were frozen in isopentane, sliced in 40 μm coronal sections in a –20°C cryostat, and examined under a light microscope. Rats with incorrect placement (fewer than 3%) were excluded from the study.

## Western blot analysis

Western blot analysis was carried out as previously reported (*Travaglia et al., 2016b*). Rats were euthanized, and their brains were rapidly removed and frozen in isopentane. dHC punches were obtained from frozen brains mounted on a cryostat using a 19-gauge neuro punch (Fine Science Tools, Foster City, CA). Samples were homogenized in ice-cold RIPA buffer (50 mM Tris base, 150 mM NaCl, 0.1% SDS, 0.5% Na-deoxycholate, 1% NP-40) containing protease and phosphatase inhibitors (0.5 mM PMSF, 2 mM DTT, 1 mM EGTA, 2 mM NaF, 1 μM microcystine, 1 mM benzamidine, 1 mM sodium orthovanadate, and commercial protease and phosphatase inhibitor cocktails [Millipore-Sigma, St. Louis,

MO]), and then centrifuged at 21,130 *g* at 4°C for 30 min. The resultant supernatant were stored at –80°C until processing. Total protein concentrations were determined using the Bio-Rad protein assay (Bio-Rad Laboratories, Hercules, CA). Equal amounts of total protein (20 μg per well) were resolved on 4–20% Criterion TGX gradient gels (Bio-Rad Laboratories, Hercules, CA) and transferred to Immobilon-FL Transfer membrane (Millipore, Billerica, MA) by electroblotting. Membranes were dried, reactivated in methanol, and washed with water before being incubated in Odyssey blocking buffer (TBS) (LI-COR Bioscience, Lincoln, NE) for 1 hr at RT. Membranes were then incubated with primary antibodies overnight at 4°C in the solution suggested by the manufacturer. Primary antibodies against the following proteins were obtained from the indicated suppliers: GR (1/2000, Abcam, cat# ab16801), GPx isozyme 1 (1/1000, Abcam, cat# ab22604), GSy (1/1000, Abcam, cat# ab133592), GCL catalytic subunit (1/1000, Novus Biologicals, cat# NBP2-45830), GCL regulatory subunit (1/1000, Abcam, cat# ab126704), NeuN (1/1000, Abcam, cat# ab104225), GFAP (1/2000, Abcam, cat# ab4674). The membranes were washed in TBS with 0.1% Tween-20 (TBST), and then incubated for 1 hr at RT with a species-appropriate fluorescently conjugated secondary antibody (goat anti-mouse IRDye 680LT [1:10,000], goat anti-rabbit IRDye 800CW [1:10,000], or donkey anti-chicken IRDye 800CW [1:10,000] from LI-COR Bioscience Lincoln, NE). Membranes were again washed in TBST and scanned using the Odyssey Infrared Imaging system (Li-Cor Bioscience). Data were quantified from pixel intensities using the Odyssey software (Image Studio 4.0). All membranes were co-stained with antibody against actin (1:10,000, Santa Cruz Biotechnology, Dallas, TX, cat# sc-47778); this signal was used as the internal loading control for all Western blots. When appropriate, membranes were stripped and re-probed with additional antibodies to detect multiple proteins on the same blot.

## GR activity assay

Rats were euthanized by decapitation at the indicated time points, and their brains were quickly removed and placed in ice-cold PBS 1×. Dorsal hippocampal samples were dissected and immediately snap-frozen in isopentane on dry ice. Samples were homogenized in ice-cold RIPA buffer, centrifuged at 21,130 *g* at 4°C for 30 min, and the resultant supernatant was collected and stored at –80°C until processing. Total protein concentrations were determined using the Bio-Rad protein assay. GR activity was measured using the Glutathione Reductase Assay kit (Millipore-Sigma, cat# GRSA-1KT). This assay is based on the reduction of GSSG by GR in presence of NADPH. In addition, 5,5'-dithiobis(2-nitrobenzoic acid) (DTNB) reacts with GSH to form 5-thio(2-nitrobenzoic acid). Therefore, the GR activity was measured by the increase in absorbance at 412 nm caused by the reduction of DTNB. In brief, 10 μL of each sample was added to each well of a 96-well plate, and then 150 μL of 2 mM GSSG, 50 μL of assay buffer (100 mM potassium phosphate, pH 7.5, with 1 mM EDTA), 75 μL of 3 mM DTNB, and finally 15 μL of 2 mM NADPH were added to each well to start the reaction. Assay buffer containing yeast GR was used as positive control. Assay buffer was used as a blank. To calculate the GR activity, the absorbance was read at 412 nm every 10 s for 2 min using the kinetic program of a spectrophotometer (Infinite M200 Plate Reader). Finally, the GR activity was normalized to the total quantity of proteins measured in each sample. GR activity was expressed in nmol/min/mg protein. All samples were run in duplicate.

## GPx activity assay

Dorsal hippocampal samples were defrosted and lysed on ice using a dounce homogenizer in 10 μL of lysis buffer (10 mM Tris, 20 mM EDTA, 320 mM sucrose, pH 7.4) per mg of tissue. Samples were then centrifuged at 21,130 *g* at 4°C for 15 min, and the supernatant were collected and transferred to a clean tube. GPx activity was measured using the Glutathione Peroxidase Assay Kit (Colorimetric) from Abcam (ab102530, Cambridge, UK) according to the manufacturer's instructions. In brief, cytosolic lysate (50 μL), standard (100 μL), GPx positive control (50 μL) or reagent control (50 μL) was added to a well in a 96-well plate. Then 40 μL of a mixture containing 33 μL of assay buffer, 3 μL of 40 mM NADPH solution, 2 μL of GR solution, and 2 μL of GSH solution were added to each well containing the samples, the positive control and the reagent control. The plate was incubated at RT for 15 min to deplete all GSSG in the samples, and the absorbance was read at 340 nm using a colorimetric microplate reader (Infinite M200 Plate Reader). Then 10 μL of cumene hydroperoxide solution were added to the sample, the positive control and the reagent control wells to start the GPx reaction, and the absorbance was measured at 340 nm every 10 s for 2 min to determine the GPx activity.

Samples producing signals greater than that of the highest standard were diluted in appropriate buffer and reanalyzed. The concentration found was then multiplied by the appropriate dilution factor. The mean absorbance value of the blank wells was substracted for each data point and the corrected absorbance values were plot for each standard as a function of the final concentration of NADPH, to construct the standard curve used to determine the values of GPx activity of the test samples. Finally, the GPx activity was normalized to the total quantity of proteins measured in each sample. All standards, controls, and samples were run in duplicate.

## GCL activity assay

GCL activity was measured as previously described (*White et al., 2003*). In brief, dorsal hippocampal samples were defrosted and lysed on ice using a dounce homogenizer in 10 µL of lysis buffer (20 mM Tris, 1 mM EDTA, 250 mM sucrose, 20 mM sodium borate, 2 mM L-serine) per mg of tissue, then centrifuged at 15,700 *g* at 4°C for 10 min. Supernatants were then redistributed to 50 µL aliquots, one for each time point, and placed at 37°C in a heat block. Fifty µL of GCL reaction buffer (400 mM Tris, 40 mM L-glutamic acid, 2 mM EDTA, 20 mM sodium borate, 2 mM L-serine, 40 mM MgCl$_2$, 40 mM ATP, pH 7.4) was added to each sample and allowed to incubate for 5 min. Reaction was started by adding 50 µL of 20 mM cysteine to samples and incubated for 20, 15, 10, and 5 min. The GCL reaction was stopped by adding 50 µL of metaphosphoric acid (2.5 g per 100 mL) to precipitate proteins, and samples were subsequently vortexed and placed on ice for 20 min. After incubation, samples were centrifuged at 580 *g* at 4°C for 5 min. Following centrifugation, 20 µL of reaction mixture was transferred to a black 96-well plate, and 180 µL of detection buffer (50 mM Tris [pH 10], 0.5 N NaOH, and 10 mM 2,3-napthalenedicarboxyaldehdye [Sigma] [v/v/v—1.4/0.2/0.2]) was added to each well. The 2,3-napthalenedicarboxyaldehdye in this mixture rapidly forms a fluorescent cyclic reaction product with the cysteine thiol and glutamyl amino groups of GSH and GCγ. The plate was left to incubate in the dark at RT for 30 min, and fluorescence intensity was measured (Ex/Em = 485/520 nm) with a microplate reader (Infinite M200 Plate Reader). GCL activity was determined by calculating the rate of fluorescence increase over time, normalized to protein content. All controls and samples were run in duplicate.

## GSy activity assay

GSy activity was measured as previously described (*Volohonsky et al., 2002*). In brief, dorsal hippocampal samples were defrosted and lysed on ice using a dounce homogenizer in 10 µL of lysis buffer (20 mM Tris, 1 mM EDTA, 250 mM sucrose, 20 mM sodium borate, 2 mM L-serine) per mg of tissue, then centrifuged at 15,700 *g* at 4°C for 10 min. Reaction mixtures contained 100 mM Tris-HCl buffer pH 8.0, 50 mM KCl, 20 mM MgCl$_2$, 2 mM EDTA, 10 mM ATP, 2.5 mM DTT, 200 µM 2-AAPA, 5 mM of gamma-glutamylcysteine, and 5 mM of glycine. Assay mixtures also contained the gamma-glutamyl transpeptidase inhibitor acivicin (500 µM – Millipore-Sigma) in order to prevent the degradation of gamma-glutamylcysteine and of the accumulating GSH. Reactions were initiated by the addition of 50 µL of samples in a final volume of 500 µL in a 96-well plate. Incubation was at 37°C and 40 µL of samples were withdrawn every 5 min up to 1 hr into 60 µL of 10% sulfosalicylic acid for the determination of GSH. The acidified samples were centrifuged at 10,000 *g* for 5 min and the supernatant was used to determine the GSH concentration using the GSH/GSSG Ratio Detection Assay Kit II (Fluorometric-Green) from Abcam (ab205811, Cambridge, UK) according to the manufacturer's instructions (see section below). Finally, the GSy activity was normalized to the total quantity of proteins measured in each sample. All samples were run in duplicate.

## GSH and GSSG level measurements

Dorsal hippocampal samples were defrosted and lysed on ice using a dounce homogenizer in 10 µL of lysis buffer (10 mM Tris, 20 mM EDTA, 320 mM sucrose, pH 7.4) per mg of tissue. Samples were then centrifuged at 21,130 *g* at 4°C for 15 min, and the supernatant was collected and transferred to a clean tube. Tissue samples may contain enzymes that can interfere with the analysis. Therefore, the enzymes were removed from the samples by using the Deproteinizing Sample Kit – TCA from Abcam (ab204708, Cambridge, UK). A portion of cytosolic lysate, taken from before deproteinization, was set aside to determine the protein concentration of each sample. GSH and total glutathione (GSH+GSSG) were measured with the GSH/GSSG Ratio Detection Assay Kit II (Fluorometric-Green) from

Abcam (ab205811, Cambridge, UK) according to the manufacturer's instructions. In brief, 96-well plates were set up in duplicate for GSH standards (50 µL), GSSG standards (50 µL), and samples (50 µL). GSH detection assay solution was prepared by diluting 100 µL of 100× Thiol Green Stock solution into 10 mL of assay buffer. For GSH detection, 50 µL of GSH detection assay solution were then added into each GSH standard and sample well to make the total assay volume 100 µL/well. Total glutathione detection assay solution was prepared by adding 5 mL of GSH detection assay solution into the GSSG Probe vial. For total glutathione detection, 50 µL of total glutathione detection assay solution were then added into each GSSG standard and sample well to make the total assay volume 100 µL/well. The 96-well plates were then left to incubate at RT for 30 min protected from light and fluorescence was measured at Ex/Em = 490/520 nm with a fluorescence microplate reader (Infinite M200 Plate Reader). Samples producing signals greater than that of the highest standard were diluted in appropriate buffer and reanalyzed. The concentration found was then multiplied by the appropriate dilution factor. Fluorescence intensity value of the blank wells was substracted for each data point and the corrected values were plot for each standard as a function of the final concentration of GSH and total glutathione (GSH+ GSSG), to construct the standard curves used to determine the GSH concentration and the total glutathione (GSH+ GSSG) concentration of the test samples. Concentration of GSSG in the test samples were calculated as: GSSG = (total glutathione – GSH)/2. Finally, the GSH and GSSG concentrations of each sample were normalized to the total quantity of proteins measured in each sample. All controls and samples were run in duplicate.

## FACS of PN17 rat hippocampal cells

Rats were euthanized by decapitation at the indicated time points. The dorsal hippocampi were rapidly dissected on wet ice in 1× PBS (pH 7.4) containing 2% fetal bovine serum. Dorsal hippocampi from three rats were pooled together per sample and processed fresh, as described by *Guez-Barber et al., 2012*. FACS was conducted at the Genomics Core facility at the Center for Genomics and Systems Biology at New York University. Neurons were tagged with a rabbit monoclonal anti-NeuN conjugated with Alexa Fluor 488 (Abcam, cat# ab190195). Astrocytes were labeled with a rat monoclonal anti-GFAP conjugated with Alexa Fluor 647 (Thermo Fisher Scientific, cat# 51-9792-82). Gating strategies were set as follows: NeuN-positive cells (NeuN+), GFAP-positive cells (GFAP+), and unlabeled counter-selected cells were homogenized in ice-cold RIPA buffer and centrifuged at 21,130 *g* at 4°C for 30 min. The resultant supernatant was collected and stored at –80°C until processing. For each sample, total protein concentration was determined using the Bio-Rad protein assay (Bio-Rad Laboratories, Hercules, CA) followed by Western blot analysis or GR activity assay, as described above.

## Statistical analysis and data visualization

The number of independent experiments carried out and the numbers of biological replicates (i.e., animals [n]) are indicated in each figure legend. No statistical method was used to predetermine sample size. The numbers of subjects used in our experiments were the minimum required to obtain statistical significance, based on our experience and in agreement with standard literature. Animals were randomly assigned to treatments or behavioral groups for all experiments.

### Metabolomic profiling

For the purpose of data visualization, the peak intensity for each metabolite was re-scaled by dividing the value for that metabolite in each sample by the median value for that metabolite across all samples, followed by log transformation (*van den Berg et al., 2006*). Missing values (less than 10%) were imputed with the minimum observed value for a given metabolite, based on the assumption that they were below the threshold of instrument detection sensitivity (*Wei et al., 2018*). Only metabolites with an assigned identity that were detected in more than 50% of the samples in a given group were included in the analysis. One- and two-way ANOVA followed by multiple comparisons tests were used to identify biochemicals that differed significantly between experimental groups. Statistical significance between two experimental groups was determined by Welch's two-sample t-test (Prism 7, GraphPad Software Inc). An estimation of the false discovery rate using q-values was calculated to take into account the multiple comparisons. p-Values < 0.05 and q-values < 0.1 were set as levels of statistical significance. q-Values were calculated according to the method described by *Storey and Tibshirani, 2003*, using Array Studio 7.2. Fold changes were calculated using the ratio of the mean

scaled intensities of two groups. Volcano plots and unsupervised HCA (using Euclidean distances) were generated in R (http://cran.r-project.org/) based on the fold change values. PCA and RF analysis (*Mitchell, 2011*) were performed using Array Studio 7.2 and the online tool MetaboAnalyst 3.6 (http://www.metaboanalyst.ca/) (*Xia and Wishart, 2016*).

### Biochemistry and behavior analyses

Analyses were performed using Prism 7 (GraphPad Software Inc). Data were analyzed by one- or two-way ANOVA for independent or repeated measures, followed by Tukey's or Bonferroni's multiple comparisons tests. Statistical significance of differences between two groups was evaluated by two-tailed Student's t-test. All analyses were two-tailed. Results were considered significant at $p < 0.05$.

The intent of this study was not to investigate sex differences; therefore, we included both female and male rats in our prepuberal groups. In the metabolomic analysis, preliminary statistical analyses of separate sex groups (n = 2–4 each sex) identified 3.08% of biochemicals that changed significantly between males and females (unpaired two-tailed Student's t-test, $p < 0.05$); because this is less than 5%, it can be considered as a random effect. At PN80, only male rats were used, hence, we cannot exclude the possibility of differential expressions of metabolites studied in adult females and males or changes related to the female reproductive cycle. Regarding the biochemistry and behavior analyses, preliminary statistical analyses of separate sex groups (n = 3–4 each sex) yielded no significant difference (unpaired two-tailed Student's t-test, $p > 0.05$), and the range of individual values was similarly distributed. Although these numbers are too low for any robust statistical analysis, we decided to group the subjects and not pursue sex-related questions.

## Acknowledgements

We thank Dr Katarzyna Broniowska (Metabolon, Inc) for technical assistance and helpful discussion and feedback. This work was supported by NIH grants R37MH065635, R01MH100822, and the DANA Foundation to CMA.

## Additional information

### Funding

| Funder | Grant reference number | Author |
|---|---|---|
| National Institute of Mental Health | MH065635 | Cristina Alberini |
| National Institute of Mental Health | MH100822 | Cristina Alberini |
| Dana Foundation | | Cristina M Alberini |

The funders had no role in study design, data collection and interpretation, or the decision to submit the work for publication.

### Author contributions

Benjamin Bessières, Conceptualization, Data curation, Funding acquisition, Methodology, Resources, Project administration, Resources, Supervision, Writing – original draft, Writing – review and editing; Emmanuel Cruz, Data curation, Methodology; Cristina M Alberini, Conceptualization, Data curation, Funding acquisition, Investigation, Methodology, Project administration, Resources, Supervision, Writing – original draft, Writing – review and editing

### Author ORCIDs

Benjamin Bessières https://orcid.org/0000-0003-4660-1170
Cristina M Alberini https://orcid.org/0000-0001-7386-0018

### Ethics

All animal procedures complied with the US National Institute of Health Guide for the Care and Use of Laboratory Animals and were approved by the New York University Animal Care Committees.

All surgeries were performed under isoflurane anesthesia and every effort was made to minimize suffering.

## Decision letter and Author response
Decision letter https://doi.org/10.7554/eLife.68590.sa1
Author response https://doi.org/10.7554/eLife.68590.sa2

## Additional files

### Supplementary files
• Transparent reporting form

### Data availability
Source data underlying Figures 1-8 and Figure 1-figure supplement 1 are provided as Source Data files. Metabolomic data is also available at the NIH Common Fund's National Metabolomics Data Repository (NMDR) website, the Metabolomics Workbench, https://www.metabolomicsworkbench. org where it has been assigned Project ID PR001118. The data can be accessed directly via its Project https://doi.org/10.21228/M86404.

The following dataset was generated:

| Author(s) | Year | Dataset title | Dataset URL | Database and Identifier |
| --- | --- | --- | --- | --- |
| Bessières B, Cruz E, Alberini CM | 2021 | Metabolic profiling of the rat hippocampus across developmental ages and after learning | https://doi.org/10.21228/M86404 | PR001118, 10.21228/M86404 |

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
