## [Editor Report]

In this study the authors examined the metabolomic profile of the rat hippocampus and report significant changes at various developmental ages at baseline and following episodic learning. Infants had the largest number of changes of hippocampal metabolites with many associated with the glutathione mediated antioxidant pathway. Infantile learning was shown to induce a rapid increase in glutathione reductase and inhibition of this enzyme impaired long term memory formation at this developmental age, but not in older animals, suggesting a key requirement for this pathway in infantile memory formation.

---

## [Decision Letter]

**Decision letter after peer review:**

Thank you for submitting your article "Metabolomic profiling reveals a differential role for hippocampal glutathione reductase in infantile memory formation" for consideration by *eLife*. Your article has been reviewed by 3 peer reviewers, and the evaluation has been overseen by a Reviewing Editor and Laura Colgin as the Senior Editor. The following individual involved in review of your submission has agreed to reveal their identity: Juan Pedro Bolaños (Reviewer #3).

Essential revisions:

1. The hippocampus is a highly heterogeneous brain structure in regards to types of neurons, glia cells, and other cells associated with blood vessels. The metabolomic profiling with the homogenates of the whole hippocampus make it difficult to interpret the data. Were the differences observed due to the differences in particular sub-regions, specific types of cells or something else? The possible input of astrocytes in the antioxidant defense during infantile learning seems to have been overlooked. For instance, in addition to the antioxidant boosts caused by NMDA extrasynaptic activity (page 20, lines 487-489; Baxter et al., 2015), in the same year (2015), another report showed that astrocytes respond to neurotransmission by boosting antioxidant support to neurons (Astrocyte NMDA receptors' activity sustains neuronal survival through a Cdk5-Nrf2 pathway. Jimenez-Blasco D, Santofimia-Castaño P, Gonzalez A, Almeida A, Bolaños JP. Cell Death Differ. 2015 Nov;22(11):1877-89. doi: 10.1038/cdd.2015.49. Epub 2015 Apr 24. PMID: 25909891). Moreover, in page 28, lines 683-685, besides the Cobley et al. (2018) paper, a subsequent study demonstrated that endogenous high mitochondrial ROS levels produced by astrocytes boost energy and redox metabolism, sustaining organismal behavior in mouse (Astrocytic mitochondrial ROS modulate brain metabolism and mouse behaviour. Vicente-Gutierrez C, Bonora N, Bobo-Jimenez V, Jimenez-Blasco D, Lopez-Fabuel I, Fernandez E, Josephine C, Bonvento G, Enriquez JA, Almeida A, Bolaños JP. Nat Metab. 2019 Feb;1(2):201-211. doi: 10.1038/s42255-018-0031-6. Epub 2019 Feb 4. PMID: 32694785). The authors should experimentally address this issue or at a minimum thoroughly discuss it since it is unclear what cell type GR activity is induced. Related to this issue, it is also unclear by what mechanism this enzyme activity is elevated after learning in a cell-free assay given that expression levels remain unaltered which should also be addressed.

2) This work consists of two major components: the metabolomic profiles of the developing hippocampus, and the metabolomic alterations caused by learning. It is not clear how these two components are linked to each other. Although reduction of GSSG was observed in both situations (in early development, and after learning in infants), it is not clear if the reduction in the two situations arise from the same or related mechanisms. The authors should address these points in the discussion.

3) Some reasoning and interpretation related to GSH, GSSG, and glutathione reductase is needed:

A) Figure 6 shows GSH/GSSG ratio, but not the level of GSH, which is more directly related to "antioxidant capacity". The data also show a continuous increase in glutathione reductase and a continuous reduction in glutathione peroxidase from PN17 to PN80, but the GSH/GSSG was not different between PN17 and PN80. An explanation of the data is needed.

B) Figure 7 shows a reduction in only GSSG but not an increase/decrease in GSH after training, suggesting that there was no increase in the conversion of GSSG into GSH, which leads to two questions: What causes GSSG to reduce? and If GSSG-to-GSH conversion did not change, why would glutathione reductase play an important role here?

(C) It is difficult to understand the data in Figure 7F. If 2-AAPA was injected into the hippocampus, when the hippocampal lysate was collected later and used for glutathione reductase activity assay, 2-AAPA may either be diluted in the assay to a negligible concentration, or remain at a high concentration which blocks glutathione reductase activity. How could it only abolish the increase in glutathione reductase activity caused by training? (D) How did injection of 2-AAPA affect the levels of GSH and GSSH?

4) Given that both cysteine levels are lower after inhibitory avoidance, and even reduced levels of glutathione show a reduction by non-adjusted p-value, the take-home message needs to be more clearly stated.

*Reviewer #1:*

Strengths:

1) The metabolomes of the hippocampus is a largely unexplored territory, especially in terms of development and learning. This study provides valuable information on these topics.

2) Identified glutathione reductase activity as a critical factor for learning in infant but not in older rats, which helps to explain the behavioral difference between infant and older rats.

3) The experiments were designed elegantly and carried out well. The data were presented clearly and convincingly. The paper was well written.

*Reviewer #2:*

The authors set out to understand the metabolic changes needed for early-life episodic memory in the hippocampus. Their main findings are that a learning paradigm of inhibitory avoidance caused a reduction in oxidized glutathione and increase in the GSH:GSSG ratio specifically in the infantile rats, associated with increased activity of glutathione reductase (GR), the enzyme which converts GSSG to GSH. Moreover pharmacological inhibition of GR impaired inhibitory avoidance at this age but not older rats. Thus, the main strength of the paper is to report a dependence on inhibitory avoidance on GR inhibition at a developmental stage when GR is activated and at which inhibitory avoidance is transient.

Main weaknesses of the paper are as follows:

i) The first 4 out of the 7 figures have little to do with the main message of the paper (as described above and corroborated by the abstract) and are instead concerned with a description of developmental changes to the hippocampal metabolome (of specialist interest).

ii) It is unclear in what cell type (neurons vs glia) GR activity is induced, and also unclear by what mechanism this enzyme activity is elevated after learning in a cell-free assay given that expression levels remain unaltered.

iii) Given that both cysteine levels are lower after inhibitory avoidance, and even reduced levels of glutathione show a reduction by non-adjusted p-value, it is not entirely clear what the take-home message is. The final sentence of the abstract "Thus, metabolomic profiling revealed that the hippocampal glutathione-mediated antioxidant pathway is differentially required for the formation of infantile memory" does not convey to me a large leap in conceptual understanding.

iv) Aside from the above, between the metabolic composition of homogenized tissue and learning/memory is a large gap, so there remain many mechanistic questions that intermediate analyses (e.g. electrophysiology, cell biology, cell-type specific KOs) would answer.

Comments for the authors:

1. For a general interest journal, it would perhaps be best to focus on the key findings that contribute to the main message of the paper.

2. Consider cell-specific KO or KD to test hypotheses.

3. Although GSSG goes down with learning at P17, so does cysteine, as GSH does also show a tendency. The effect on GSH in particular needs more investigation. I'm not sure whether there is an increase or decrease in available GSH after learning, which is fundamental.

4. Westerns for glutathione peroxidase-there are many such Gpx enzymes-do they mean Gpx1?

5. Glutathione synthetase westerns-although there are differences observed the authors need to make the case that this could be rate-determining. Many studies have found that gclc activity is rate-determining.

6. The effect of the irreversible GR inhibitor drug AAPA is surprising-it does not block GR activity as one would expect (7F) but simply reduces it to the level of untrained mice. This is critical to the paper and I am unclear how this effect is modest.

*Reviewer #3:*

This is an original work showing that boosted hippocampal glutathione reductase enzymatic activity is required for learning, selectively during the infantile period of the rat. The study is elegantly designed and is experimentally very robust, giving rise to an important piece of work in which the main conclusions are supported by the data. There are previous reports linking energy and redox metabolism in development and aging, but this work adds novel insight on the importance of these aspects of metabolism in memory formation, leading to the elucidation of a specific pathway that is compatible with what is already known in the field of redox biology and learning. The study is comprehensive, and the methodological approaches are diverse and elegant. This reviewer has no major comments, besides some suggestions aimed to improve the fitting of their results in the context of recent, previous reports that may add extra value to the discussion.

1. The authors might have inadvertently overlooked a previous work demonstrating that the hippocampal neurons of adult (9 months-old) mouse require a large pool of glutathione to sustain cognitive functions (Hippocampal neurons require a large pool of glutathione to sustain dendrite integrity and cognitive function. Fernandez-Fernandez S, Bobo-Jimenez V, Requejo-Aguilar R, Gonzalez-Fernandez S, Resch M, Carabias-Carrasco M, Ros J, Almeida A, Bolaños JP. Redox Biol. 2018 Oct;19:52-61. doi: 10.1016/j.redox.2018.08.003. Epub 2018 Aug 7. PMID: 30107295). The authors should consider to discuss this work in the context of their own findings, given that a high GSH requirement in adulthood is compatible -and might even explain- why in adulthood the GSH-mediated antioxidant capacity in the adult hippocampus is higher than in the infantile and juvenile ones (Figure 6 and the text in discussion related to this finding).

2. On pages 18 (lines 433 to 440) and 19 (lines 441-450), the authors discuss why there should be such a high antioxidant demand following learning in infancy. They speculate that higher level of oxidative stress during infantile learning would be due to a high metabolic activity. However, this reviewer does not understand why an increased metabolic activity should increase ROS production. In this context, an increase in the bioenergetic efficiency of mitochondria (i.e., high metabolic activity) decreases ROS production by mitochondria (Complex I assembly into supercomplexes determines differential mitochondrial ROS production in neurons and astrocytes. Lopez-Fabuel I, Le Douce J, Logan A, James AM, Bonvento G, Murphy MP, Almeida A, Bolaños JP. Proc Natl Acad Sci U S A. 2016 Nov 15;113(46):13063-13068. doi: 10.1073/pnas.1613701113. Epub 2016 Oct 31. PMID: 27799543), which would argue against their rationale. May the authors consider debating about this?

3. The authors have studied the hippocampus, which besides neurons obviously contains a large proportion of astrocytes (e.g., Hippocampal volume and total cell numbers in major depressive disorder. Cobb JA, Simpson J, Mahajan GJ, Overholser JC, Jurjus GJ, Dieter L, Herbst N, May W, Rajkowska G, Stockmeier CA. J Psychiatr Res. 2013 Mar;47(3):299-306. doi: 10.1016/j.jpsychires.2012.10.020. Epub 2012 Nov 30. PMID: 23201228) and other cell types. However, the possible input of astrocytes in the antioxidant defense during infantile learning seems to have been overlooked. For instance, in addition to the antioxidant boos caused by NMDA extrasynaptic actiuvity (page 20, lines 487-489; Baxter et al., 2015), in the same year (2015), another report showed that astrocytes respond to neurotransmission by boosting antioxidant support to neurons (Astrocyte NMDA receptors' activity sustains neuronal survival through a Cdk5-Nrf2 pathway. Jimenez-Blasco D, Santofimia-Castaño P, Gonzalez A, Almeida A, Bolaños JP. Cell Death Differ. 2015 Nov;22(11):1877-89. doi: 10.1038/cdd.2015.49. Epub 2015 Apr 24. PMID: 25909891). The authors should consider bringing astrocytes in their discussion. Moreover, in page 28, lines 683-685, besides the Cobley et al. (2018) paper, a subsequent study demonstrated that endogenous high mitochondrial ROS levels produced by astrocytes boost energy and redox metabolism, sustaining organismal behavior in mouse (Astrocytic mitochondrial ROS modulate brain metabolism and mouse behaviour. Vicente-Gutierrez C, Bonora N, Bobo-Jimenez V, Jimenez-Blasco D, Lopez-Fabuel I, Fernandez E, Josephine C, Bonvento G, Enriquez JA, Almeida A, Bolaños JP. Nat Metab. 2019 Feb;1(2):201-211. doi: 10.1038/s42255-018-0031-6. Epub 2019 Feb 4. PMID: 32694785). This work also highlights an important role for astrocytes in providing antioxidant protection to neurons. Therefore, the authors may find these works relevant in the context of that part of their discussion, likely providing a potential role for astrocytes in their main message.

---

## [Author Response]

Essential revisions:1. The hippocampus is a highly heterogeneous brain structure in regards to types of neurons, glia cells, and other cells associated with blood vessels. The metabolomic profiling with the homogenates of the whole hippocampus make it difficult to interpret the data. Were the differences observed due to the differences in particular sub-regions, specific types of cells or something else? The possible input of astrocytes in the antioxidant defense during infantile learning seems to have been overlooked. For instance, in addition to the antioxidant boosts caused by NMDA extrasynaptic activity (page 20, lines 487-489; Baxter et al., 2015), in the same year (2015), another report showed that astrocytes respond to neurotransmission by boosting antioxidant support to neurons (Astrocyte NMDA receptors' activity sustains neuronal survival through a Cdk5-Nrf2 pathway. Jimenez-Blasco D, Santofimia-Castaño P, Gonzalez A, Almeida A, Bolaños JP. Cell Death Differ. 2015 Nov;22(11):1877-89. doi: 10.1038/cdd.2015.49. Epub 2015 Apr 24. PMID: 25909891). Moreover, in page 28, lines 683-685, besides the Cobley et al. (2018) paper, a subsequent study demonstrated that endogenous high mitochondrial ROS levels produced by astrocytes boost energy and redox metabolism, sustaining organismal behavior in mouse (Astrocytic mitochondrial ROS modulate brain metabolism and mouse behaviour. Vicente-Gutierrez C, Bonora N, Bobo-Jimenez V, Jimenez-Blasco D, Lopez-Fabuel I, Fernandez E, Josephine C, Bonvento G, Enriquez JA, Almeida A, Bolaños JP. Nat Metab. 2019 Feb;1(2):201-211. doi: 10.1038/s42255-018-0031-6. Epub 2019 Feb 4. PMID: 32694785). The authors should experimentally address this issue or at a minimum thoroughly discuss it since it is unclear what cell type GR activity is induced. Related to this issue, it is also unclear by what mechanism this enzyme activity is elevated after learning in a cell-free assay given that expression levels remain unaltered which should also be addressed.

We thank the Reviewer for these great comments, which led us to better clarify the scope of our study and to extend our assessments to neuron vs. astrocyte selective regulations.

As detailed in the revised paper, we aimed at obtaining a metabolomic investigation at the whole hippocampus level because the goal was to gain a comprehensive knowledge of the hippocampal metabolome changes over ages and in response to learning. We think that identifying the cell-type- and subregion-specific metabolomes is a future goal, which will be guided by our current comprehensive data. Dissecting all or most cell-type and subregion metabolomes would have multiplied our metabolomic experiments to a number that would be unfeasible for a single manuscript. Furthermore, we would have missed the comprehensive profile of changes, which we believe is very valuable for designing future studies and analyses. Therefore, as suggested by the Reviewer, we now better detail the intent of the experimental approach and discuss that future goals should include the investigation of different cell types and hippocampal subregions (pages 16 and 20).

We also added discussion points on the possible role of astrocytes in the antioxidant defense during infantile memory formation. Taking into account recent studies, including the ones mentioned by the reviewers, we added the following point of discussion on page 16:

“As our biochemical analyses were obtained from hippocampal extracts, they did not dissect whether the learning-induced metabolomic changes and increase in GR activity are distinctively regulated within the different hippocampal subregions or cell types. […] Yet, a possible role of astrocytes in the antioxidant defense during infantile memory formation remains to be determined.”

Finally, we investigated whether the learning-induced increase in GR activity is differentially regulated in neurons and in astrocytes. We measured GR activity in FACS-sorted neurons, astrocytes and unlabeled counter-selected brain cell-types (i.e. oligodendrocytes, microglia, vascular cells etc.) from the dorsal hippocampus of PN17 untrained and trained rats euthanized 15 min post-training, a timepoint at which we found a significant induction of GR activity in the total dHC extract. We now report that infant learning significantly increases GR activity in neurons but not in astrocytes or the remaining brain cell fraction (new Figure 7g).

Regarding the possible mechanisms by which GR activity increases after learning despite its level remains unaltered, we now have added the following discussion point, in which we propose explanations (see page 15):

“The mechanisms by which GR activity increases following learning without a change in GR expression level itself are not known. […] Despite GR exhibits several phosphorylation, acetylation and ubiquitylation sites (Deponte, 2013), very little is known about the relevance of these posttranslational modifications in modulating GR activity, and the effects of learning on GR posttranslational modifications or allosteric modulations remain to be understood.”

2) This work consists of two major components: the metabolomic profiles of the developing hippocampus, and the metabolomic alterations caused by learning. It is not clear how these two components are linked to each other. Although reduction of GSSG was observed in both situations (in early development, and after learning in infants), it is not clear if the reduction in the two situations arise from the same or related mechanisms. The authors should address these points in the discussion.

We followed the reviewer’s suggestion and added a discussion point on this interesting question (see page 12):

“Future investigation shall analyze the relationship between developmental and learning-induced changes. […] Given our previously studies showing that biological and functional development of the hippocampus does not progress by default, but it is instructed by learning experiences especially in infancy (Bessières et al. 2020), we suggest that at least several changes that occur over development will be also found to be associated with learning.”

3) Some reasoning and interpretation related to GSH, GSSG, and glutathione reductase is needed:A) Figure 6 shows GSH/GSSG ratio, but not the level of GSH, which is more directly related to "antioxidant capacity". The data also show a continuous increase in glutathione reductase and a continuous reduction in glutathione peroxidase from PN17 to PN80, but the GSH/GSSG was not different between PN17 and PN80. An explanation of the data is needed.

We thank the Reviewer for pointing this out. We updated Figure 6, and now show the levels of GSH and GSSG at the various ages, in addition to the GSH/GSSG ratios (Figure 6b). These data revealed that the level of GSH was significantly higher at PN24 compared to PN17 and PN80, whereas GSSG levels of PN17 and PN24 were significantly higher relative to that of PN80. As a result, the GSH/GSSG mean ratio was significantly higher at PN24 compared to PN80 and slightly but not significantly lower at PN17 relative to PN80. Despite the lack of significant decrease in the mean ratio of PN17 over PN80, there was a broader distribution of GSSG individual values at PN17 compared to PN80, suggesting that the regulation of the glutathione cycle might be different in the infant hippocampus. Overall, these results suggest a higher antioxidant capacity at PN24.

Furthermore, we also now report the assessment of the enzymatic activities of glutathione reductase (GR), glutathione peroxidase (GPx), glutamate-cysteine ligase (GCL), and glutathione synthetase (GSy) across developmental ages. These new data, presented Figure 6d, show that the levels of GR and GPx activity were significantly higher at PN17 and PN24 compared to PN80, suggesting that there is a higher rate of glutathione cycle during early developmental ages compared to adulthood.

We also found that the level of activity of GCL dramatically increased between PN17 and PN24 and then decreased in adulthood at a level that however remained significantly elevated compared to PN17 (Figure 6d). Likewise, the activity of GSy was significantly higher at PN24 and PN80 compared to PN17. These results, in line with the protein expression levels (Figure 6c), suggest a reduced de novo GSH synthesis in infancy compared to juvenile and adults, which could be explained by a lower level of the rate-limiting GSH precursor cysteine and/or of GCLM at PN17 (Figure 6b,c).

Collectively, these results suggest that the hippocampal glutathione-mediated antioxidant defense capacity is lower in the infant than in the juvenile and adult hippocampus. We have added this new explanation and a discussion point on pages 14-15.

B) Figure 7 shows a reduction in only GSSG but not an increase/decrease in GSH after training, suggesting that there was no increase in the conversion of GSSG into GSH, which leads to two questions: What causes GSSG to reduce? and If GSSG-to-GSH conversion did not change, why would glutathione reductase play an important role here?

The reviewer raised an important point regarding the absence of GSH variation at 1h after PN17 learning, despite there is an increase in GR activity, which in principle should increase the conversion of GSSG into GSH. To address this point we dissected how infant learning affects the activity of the critical enzymes involved in glutathione oxidation/reduction cycle (i.e. Gpx and GR) and biosynthesis (i.e. GCL and GSy), as well as the GSH and GSSG levels. Toward this end, we performed time course analyses of the activity of the four enzymes and of the levels of the two metabolites in the dorsal hippocampus following infant learning. We found that, following training, GR activity increased significantly at 15 min after training and remained elevated at 1h and 24h before returning to baseline by 7 days after training (new Figure 7c). The activities of all the other enzymes, i.e., GPx, GLC and GSy remained unchanged. These results suggest that learning at PN17 induces an increase in the regeneration of GSH from GSSG, while the conversion of GSH into GSSG and the biosynthesis of GSH remain unaltered.

We also found that infant learning leads to a significant increase in GSH level paralleled by a decrease in GSSG at 15 min after training. Notably, the level of GSH returns to baseline at 1h and 24h after training, whereas at the same time points, the levels of GSSG remained significantly lower than those observed in untrained rats (new Figure 8e-g). This timecourse is consistent with our metabolomic data showing that GSSG significantly decreased while GSH was unchanged relative to untrained animals at 1 hour after training (Figure 7a).

The reason why GSH rapidly returned to basal level (at the 1h time point) while GSSG level remains reduced and GR activity elevated compared to untrained rats remains unknown. We suggest that GSH is rapidly consumed between 15 min and 1h to maintain the redox balance or to respond to additional unknown metabolic requirements following learning.

Furthermore, we found that the learning-induced changes in hippocampal GSH and GSSG levels were prevented by bilateral injection of the GR inhibitor 2-AAPA (100 μM) in the dorsal hippocampus, 15 min before training (new Figure 8a and 8e-g). These results confirmed that the learning-induced GSH increase and GSSG decrease was mediated by an increase in GR activity following learning.

In sum we added several sets of data and relative discussion points (see page 18) showing the critical role played by GR activity to modulate GSH and GSSG levels evoked by learning.

(C) It is difficult to understand the data in Figure 7F. If 2-AAPA was injected into the hippocampus, when the hippocampal lysate was collected later and used for glutathione reductase activity assay, 2-AAPA may either be diluted in the assay to a negligible concentration, or remain at a high concentration which blocks glutathione reductase activity. How could it only abolish the increase in glutathione reductase activity caused by training?

To address the Reviewer important point, we added the assessment of a higher 2-AAPA concentration. As shown by our previous data and mentioned by the reviewer, 2-AAPA injected at 100 μM in the dHC 15 min before training at PN17 prevented the learning-induced increase in GR activity. We hypothesized that at this concentration, 2-AAPA partially inhibits GR. When we injected before training a higher dose of 2-AAPA (i.e., 200 μM) in the dHC, we found that GR activity is decreased by more than 75% (new Figure 8a). Therefore 100μM of 2-AAPA indeed was partially inhibiting GR allowing us to block the increase in GR activity induced by learning. In sum, the partial effect of 100 μM concentration was not due to a temporal effect of 2-AAPA but to a partial inhibition. Our result also show that the injection of 2-AAPA at the time of training durably alter the ability of learning to increase GR catalytic activity, which is in line with the *in vitro* observations that the GR inhibition by 2-AAPA is irreversible (Seefeldt et al. 2009).

(D) How did injection of 2-AAPA affect the levels of GSH and GSSH?

Please see response to the comment 3 (B).

4) Given that both cysteine levels are lower after inhibitory avoidance, and even reduced levels of glutathione show a reduction by non-adjusted p-value, the take-home message needs to be more clearly stated.

We thank the reviewer for this comment. Regarding the small reduction of GSH following training in infancy, please see our response to comment 3 (B) and our new data showing that GSH significantly increases at 15 min after training (new Figure 8e). Regarding the significant decrease in cysteine level following IA training, we have added the following discussion point to address this question (see page 10).

“Despite cysteine decreases at 1 h following learning, at this timepoint we did not detect significant change in GSH level (Figure 5a and d). Therefore, we speculate that the decrease in cysteine may not be linked to GSH biosynthesis but rather serves as a substrate for de novo protein synthesis, a process known to be required for long-term memory formation (Costa-Mattioli et al., 2009).”

Reviewer #2:[…] Comments for the authors:1. For a general interest journal, it would perhaps be best to focus on the key findings that contribute to the main message of the paper.2. Consider cell-specific KO or KD to test hypotheses.

We appreciate the reviewer’s comment and now better clarified the rationale of our experimental design (see page 16 and 20).

The main goal of this study was to provide a comprehensive analysis of the metabolomic changes occurring in the rat hippocampus across post-natal developmental ages and after learning, rather than focusing on specific cell types. To our knowledge, this topic has not been explored yet. Dissecting metabolomics of specific cell types is a different and, in our view, subsequent question. The technique of dissecting certain cell types may also lead to the loss of certain metabolites, which we wanted to avoid in this initial metabolomic study. A second question is: which cell types to choose? In the hippocampus we have neurons (to be divided in a variety of subpopulations), astrocytes, oligodendrogytes, microglia, endothelial cells, pericytes,… hence many different cell types. Why favoring neurons (which ones?) and astrocytes? Our intent was to not focus on 1 or 2 cell types but to get a comprehensive knowledge of the metabolomic changes of the region. We agree that diving into additional levels of cell type and subregion analyses is also very interesting and we see these questions as subsequent to this study.

Specifically, our goal was to define the global metabolomic changes driven by (1) development, which per se is important and (2) learning at different ages. Without defining the metabolomics of different ages, we could not have obtained meaningful profiling underlying learning over developmental ages. From these analyses we identified a few metabolic pathways significantly and differentially regulated by learning at different ages. Then we selected one of them that emerged as regulated by learning at PN17, the glutathione-mediated antioxidant pathway, for a more in-depth investigation of causality in memory formation. However, the intent of this study was not to investigate the detailed mechanistic aspects of the changes revealed by our metabolomic analysis, which may be the object of further investigations.

3. Although GSSG goes down with learning at P17, so does cysteine, as GSH does also show a tendency. The effect on GSH in particular needs more investigation. I'm not sure whether there is an increase or decrease in available GSH after learning, which is fundamental.

We addressed this issue by adding several new data sets as described in the response to the comments 3 (B) and 4, above.

4. Westerns for glutathione peroxidase-there are many such Gpx enzymes-do they mean Gpx1?

We are grateful for asking to clarify this point. The glutathione peroxidase analyzed in our experiments is the isozyme GPx1. We now specify the correct nomenclature in the manuscript.

5. Glutathione synthetase westerns-although there are differences observed the authors need to make the case that this could be rate-determining. Many studies have found that gclc activity is rate-determining.

There are no differences at 1h and 24h after training in the activity of GCL and GSy, compared to untrained rats. Although some of these enzymes can be rate limiting in some conditions, our two timepoints after training do not reveal changes in the activity of these enzymes. These data do not exclude that activation of the enzymes are regulated (not their expression level) or that there are changes at other timepoints. Additional investigations should be the subject of future studies. We also emphasized that the activity of GCL is rate-limiting in the biosynthesis of GSH (page 13).

6. The effect of the irreversible GR inhibitor drug AAPA is surprising-it does not block GR activity as one would expect (7F) but simply reduces it to the level of untrained mice. This is critical to the paper and I am unclear how this effect is modest.

Please see response to comment 3 (C).

Reviewer #3:This is an original work showing that boosted hippocampal glutathione reductase enzymatic activity is required for learning, selectively during the infantile period of the rat. The study is elegantly designed and is experimentally very robust, giving rise to an important piece of work in which the main conclusions are supported by the data. There are previous reports linking energy and redox metabolism in development and aging, but this work adds novel insight on the importance of these aspects of metabolism in memory formation, leading to the elucidation of a specific pathway that is compatible with what is already known in the field of redox biology and learning. The study is comprehensive, and the methodological approaches are diverse and elegant. This reviewer has no major comments, besides some suggestions aimed to improve the fitting of their results in the context of recent, previous reports that may add extra value to the discussion.1. The authors might have inadvertently overlooked a previous work demonstrating that the hippocampal neurons of adult (9 months-old) mouse require a large pool of glutathione to sustain cognitive functions (Hippocampal neurons require a large pool of glutathione to sustain dendrite integrity and cognitive function. Fernandez-Fernandez S, Bobo-Jimenez V, Requejo-Aguilar R, Gonzalez-Fernandez S, Resch M, Carabias-Carrasco M, Ros J, Almeida A, Bolaños JP. Redox Biol. 2018 Oct;19:52-61. doi: 10.1016/j.redox.2018.08.003. Epub 2018 Aug 7. PMID: 30107295). The authors should consider to discuss this work in the context of their own findings, given that a high GSH requirement in adulthood is compatible -and might even explain- why in adulthood the GSH-mediated antioxidant capacity in the adult hippocampus is higher than in the infantile and juvenile ones (Figure 6 and the text in discussion related to this finding).

We thank the reviewer for pointing this out and apologize to have inadvertently overlooked that manuscript. We now added the following point of discussion related to the important reference (see page 19):

“Finally, our results showing that inhibiting GR at the time of learning in juvenile or adult rats has no effect on memory performances could be explained by a higher level of GCL and GSy activities (as shown Figure 6) and/or of differential antioxidant mechanisms present in the adult and juvenile hippocampi. Future investigations will determine whether learning in juveniles and adults affects the activity of GCL and GSy and if these activities are required for memory formation. In a previous study, it has been shown that a moderate decrease of neuronal GSH in CA1 area of the hippocampus of adult mice by GCLC knockdown leads to dendrite disruption and cognitive impairment, revealing that hippocampal neurons of adult mouse require a large pool of glutathione to sustain cognitive functions (Fernandez-Fernandez et al., 2018).”

2. On pages 18 (lines 433 to 440) and 19 (lines 441-450), the authors discuss why there should be such a high antioxidant demand following learning in infancy. They speculate that higher level of oxidative stress during infantile learning would be due to a high metabolic activity. However, this reviewer does not understand why an increased metabolic activity should increase ROS production. In this context, an increase in the bioenergetic efficiency of mitochondria (i.e., high metabolic activity) decreases ROS production by mitochondria (Complex I assembly into supercomplexes determines differential mitochondrial ROS production in neurons and astrocytes. Lopez-Fabuel I, Le Douce J, Logan A, James AM, Bonvento G, Murphy MP, Almeida A, Bolaños JP. Proc Natl Acad Sci U S A. 2016 Nov 15;113(46):13063-13068. doi: 10.1073/pnas.1613701113. Epub 2016 Oct 31. PMID: 27799543), which would argue against their rationale. May the authors consider debating about this?

We thank the Reviewer for raising this interesting point. We now added discussion points that better elaborate on the possible reasons for why learning in infancy is associated with a high antioxidant demand. We also clarified our hypothesis regarding the link between the level of metabolic activity and ROS production. Specifically, we added (page 10):

“Indeed, neuronal electrical activity is known to raise ATP demands, which must be met by increased metabolic activity, particularly oxidative phosphorylation (Harris et al. 2012; Fernandez-Fernandez et al. 2012; Bolaños 2016), which results in an increased production of reactive oxygen species (ROS) by the mitochondrial respiratory chain, potentially leading to oxidative stress (Hongpaisan et al. 2004; Brennan et al. 2009; Frisard and Ravussin, 2006; Quijano et al. 2016). […] Future studies should assess the contributions of astrocytic-neuronal metabolic coupling associated to hippocampus-dependent memory formation in infants relative to adults.”

3. The authors have studied the hippocampus, which besides neurons obviously contains a large proportion of astrocytes (e.g., Hippocampal volume and total cell numbers in major depressive disorder. Cobb JA, Simpson J, Mahajan GJ, Overholser JC, Jurjus GJ, Dieter L, Herbst N, May W, Rajkowska G, Stockmeier CA. J Psychiatr Res. 2013 Mar;47(3):299-306. doi: 10.1016/j.jpsychires.2012.10.020. Epub 2012 Nov 30. PMID: 23201228) and other cell types. However, the possible input of astrocytes in the antioxidant defense during infantile learning seems to have been overlooked. For instance, in addition to the antioxidant boos caused by NMDA extrasynaptic activity (page 20, lines 487-489; Baxter et al., 2015), in the same year (2015), another report showed that astrocytes respond to neurotransmission by boosting antioxidant support to neurons (Astrocyte NMDA receptors' activity sustains neuronal survival through a Cdk5-Nrf2 pathway. Jimenez-Blasco D, Santofimia-Castaño P, Gonzalez A, Almeida A, Bolaños JP. Cell Death Differ. 2015 Nov;22(11):1877-89. doi: 10.1038/cdd.2015.49. Epub 2015 Apr 24. PMID: 25909891). The authors should consider bringing astrocytes in their discussion. Moreover, in page 28, lines 683-685, besides the Cobley et al. (2018) paper, a subsequent study demonstrated that endogenous high mitochondrial ROS levels produced by astrocytes boost energy and redox metabolism, sustaining organismal behavior in mouse (Astrocytic mitochondrial ROS modulate brain metabolism and mouse behaviour. Vicente-Gutierrez C, Bonora N, Bobo-Jimenez V, Jimenez-Blasco D, Lopez-Fabuel I, Fernandez E, Josephine C, Bonvento G, Enriquez JA, Almeida A, Bolaños JP. Nat Metab. 2019 Feb;1(2):201-211. doi: 10.1038/s42255-018-0031-6. Epub 2019 Feb 4. PMID: 32694785). This work also highlights an important role for astrocytes in providing antioxidant protection to neurons. Therefore, the authors may find these works relevant in the context of that part of their discussion, likely providing a potential role for astrocytes in their main message.

Please see response to the comment #1 of the Essential revisions.